# Discovery of Options via Meta-Learned Subgoals

**Vivek Veeriah**[*]
University of Michigan

**Tom Zahavy**
DeepMind

**Matteo Hessel**
DeepMind

**Zhongwen Xu**[†]
DeepMind

**Junhyuk Oh**
DeepMind

**Iurii Kemaev**
DeepMind

**Hado van Hasselt**
DeepMind

**David Silver**
DeepMind

**Satinder Singh**
University of Michigan, DeepMind

## Abstract

Temporal abstractions in the form of options have been shown to help reinforcement learning (RL) agents learn faster. However, despite prior work on this topic, the problem of discovering options through interaction with an environment remains a challenge. In this paper, we introduce a novel meta-gradient approach for discovering useful options in multi-task RL environments. Our approach is based on a manager-worker decomposition of the RL agent, in which a manager maximises rewards from the environment by learning a task-dependent policy over both a set of task-independent discovered-options and primitive actions. The option-reward and termination functions that define a subgoal for each option are parameterised as neural networks and trained via meta-gradients to maximise their usefulness. Empirical analysis on gridworld and DeepMind Lab tasks show that: (1) our approach can discover meaningful and diverse temporally-extended options in multi-task RL domains, (2) the discovered options are frequently used by the agent while learning to solve the training tasks, and (3) that the discovered options help a randomly initialised manager learn faster in completely new tasks.

Reinforcement learning (RL) problems involve learning about temporally-extended actions at multiple time scales. In RL, the *options* framework (Sutton et al., 1999) provides a well-defined formalisation for the notion of temporally-extended actions. Options that achieve specific subgoals can be useful in reinforcement learning (RL) in at least two ways: in model-based RL, they provide faster rates of convergence through longer-backups of value functions within planning updates (Silver & Ciosek, 2012; Mann & Mannor, 2014; Brunskill & Li, 2014), while in model-free RL, temporally-extended actions commit agents to intentional multi-step behaviours, which can translate into better exploration (Machado & Bowling, 2016; Nachum et al., 2019; Osband et al., 2019).

We consider a scenario where an agent learns to solve a distribution over tasks. In such cases having carefully designed temporal abstractions can greatly reduce the overall sample complexity of learning for an RL agent that is trying to master those tasks. The agent can produce faster learning mainly by reusing those abstractions across multiple tasks (Sutton et al., 1999; Solway et al., 2014). Many recent approaches have empirically validated this by demonstrating that hand-designed temporal abstractions can often lead to improved learning performance on a variety of challenging multi-task RL domains (Imazeki & Maeno, 2003; Kulkarni et al., 2016; Nachum et al., 2018; Riedmiller et al., 2018). However, if the agent has abstractions that are not useful to a downstream task, it can significantly hurt the performance by making exploration harder (Jong et al., 2008), emphasising the challenge involved in carefully hand-designing abstractions that are useful *in general*, across many domains and tasks. Thus, the automated discovery of temporal abstractions from experience without extensive domain-specific knowledge remains an important open problem for reducing sample complexity in RL.

---

[*]Corresponding Author. Email: `vveeriah@umich.edu`

[†]Now at Sea AI Lab

35th Conference on Neural Information Processing Systems (NeurIPS 2021).

The *main contribution* of this work is our hierarchical agent architecture and an associated meta-gradient algorithm to discover temporally-extended actions in the form of options that can be reused across many tasks. Previously, meta-gradients have been successfully used for learning hyperparameters (Xu et al., 2018; Zahavy et al., 2020), intrinsic rewards (Zheng et al., 2018, 2019; Rajendran et al., 2019), and auxiliary tasks (Veeriah et al., 2019). Our work is the first to demonstrate that they can successfully learn rich parameterisations of temporal abstractions. Our starting point is the following hypothesis: If we could *discover* temporal abstractions useful across *many training tasks*, they would capture regularities across those task environments and have a higher likelihood of being useful and reusable in new, *previously unseen tasks.*

To discover temporal abstractions useful across tasks, our system flexibly defines *task-independent* subgoals for options through separate discovered *rewards* and *terminations*, different for each option. We employ meta-gradients to discover the parameters of such option-rewards and terminations based on their utility across the many training tasks, so that the set of induced options is useful to a hierarchical agent trying to master all training tasks. The meta-gradient approach operates by evaluating a *change* in the options, caused via changes to the option-rewards and terminations, w.r.to the hierarchical agent's performance on samples drawn from many tasks; then computes and uses the gradients from this evaluation to *discover* the option-rewards and terminations. This differs significantly from the previous multi-task option discovery approaches (Bacon et al., 2017; Frans et al., 2017), where all options directly optimise the same (main task) reward, which may be insufficient to discover reusable, task-independent options.

We evaluate the proposed approach empirically in two multi-task RL settings based on an illustrative gridworld and on 3-dimensional first-person task suites from DeepMind Lab (Beattie et al., 2016). For each of these, we perform three types of analysis: (1) we qualitatively demonstrate that our approach indeed produces meaningful temporally-extended options; (2) we quantitatively show that our discovered options are extensively used by the agent while learning to solve training tasks; (3) we show that our discovered options support faster learning, i.e., transfer better, in test tasks, compared to options discovered by two strong hierarchical baselines (i.e., MLSH (Frans et al., 2017) & Option-Critic (Harb et al., 2018)).

# 1 MODAC: Meta-gradients for Option Discovery using Actor-Critic

**Why Meta-gradients?** As described in the previous section, the motivation behind our work is to discover temporal abstractions in the form of options that are *generally* useful across many training tasks; and to leverage them to allow effective transfer of acquired skills to new tasks. This is based on our hypothesis that options, if useful across many tasks, capture intrinsic properties about those tasks that could lead to better transfer to unseen tasks. For useful options to be discovered, an option-based agent needs to evaluate whether *a change* in a given option is useful. Essentially, the agent needs to compute the gradient of future performance with respect to the parameterisation of each option, while the option-based behaviour is itself adapted by the conventional RL gradients; this requires the computation of a gradient through a gradient, and thus meta-gradients is used as the mechanism for driving option discovery. We make this idea concrete in our learning agent MODAC (which stands for **M**eta-gradients for **O**ption **D**iscovery using **A**ctor-**C**ritic).

**Background on Options:** Options are closed-loop behaviours over extended periods of time. An option is formally defined by specifying an *initiation set* (states where the option may be invoked), an *option-policy* (maps states to actions), and a *termination* function (maps states to probability of terminating execution of the option). The option-policy may be defined implicitly as the policy that maximises an *option-reward* function describing the subgoal (or *intention*) for that option.

## 1.1 Agent Architecture

We base our agent's architecture on the standard hierarchical agent from Sutton et al. (1999), where a manager chooses among both primitive actions[3] and temporally-extended options; and extend it to the multi-task setup. The agent (shown in Fig. 1) follows a *call-and-return* option execution-model and consists of the following four modules:

---

[3]Primitive actions are a special case of options that terminate after one step. See Sec. 3.1, pg. 194 of Sutton et al. (1999).

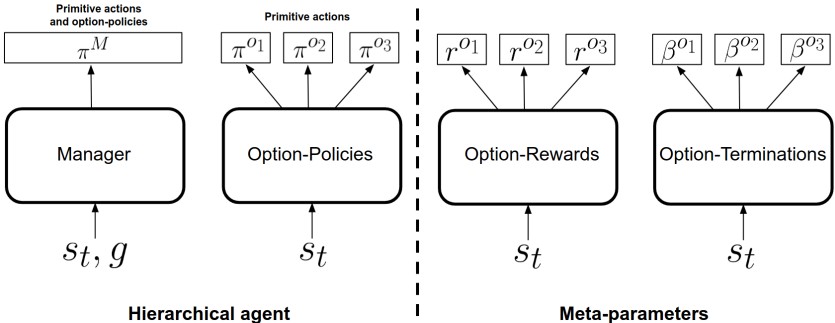

Figure 1: MODAC: features four networks, namely: the manager-policy, option-policies, option-rewards, and option-terminations. Manager and option-policies are trained via (direct) RL. Option-rewards and terminations define the semantics for the option-policies and are discovered (indirectly) using meta-gradient RL. More details in Sec. 1.1.

A **Manager** network parameterised by $\theta^M$ implements task-conditional policy $\pi^M$. It maps a sequence of observations (henceforth the state $s_t \in \mathcal{S}$) and a task encoding $g \in \mathcal{G}$ to the union of the set of option-policies $\mathcal{O}$ and the set of primitive actions $\mathcal{A}$, i.e., $\pi^M : \mathcal{S} \times \mathcal{G} \rightarrow \{\mathcal{O}, \mathcal{A}\}$. We denote a sample from $\pi^M$ as $o \sim \pi^M(s, g)$. The Manager is trained to maximise the task reward.

The **Option-policy** network represents policies for each discovered option in our hierarchical agent. It takes state as input and has multiple output head that each produce a distribution over primitive actions: the $i^{th}$ head's output represents the option-policy $\pi^{o_i} : \mathcal{S} \rightarrow \mathcal{A}$). $\theta^{o_i}$ denotes the parameters of $\pi^{o_i}$. Each option-policy learns to maximise the corresponding option-rewards produced from the option-reward network described next.

The **Option-reward** network defines the subgoals/intentions for the option-policies. It takes state as input and outputs a scalar reward, one per option-policy, for each primitive action $a \in \mathcal{A}$ (the $i^{th}$ option-reward head's output is $r^{o_i} : \mathcal{S} \rightarrow \mathbb{R}^{|\mathcal{A}|}$). $\eta^{r^{o_i}}$ denotes the parameters of $r^{o_i}$.

**Option-termination** network defines the termination function for the option-policies and has an identical structure to the option-reward network: it also takes state as input and outputs a scalar value $\beta^{o_i} : \mathcal{S} \rightarrow \{0, 1\}$ for each option-policy, interpreted as the probability of termination for the associated option. $\eta^{\beta^{o_i}}$ denotes the parameters of $\beta^{o_i}$.

Each module in the agent closely follows the options framework: the manager corresponds to the *policy-over-options* from the options framework as it learns a mapping from states to options. Each option is defined by its associated option-policy and termination defined by the respective networks (initiation set for each option is softly induced by the manager's policy). Option-reward along with their termination defines an option's subgoal.

Note, among the four modules, only the manager gets the task goal as input while those corresponding to options (i.e., option-policy, reward and termination) do not. This architecturally enforces our objective of discovering task-independent options that are useful across multiple training tasks, and also supports their transfer to new, previously unseen tasks. We emphasise that it is possible for MODAC to solve each task optimally as the manager gets task as input and can select primitive actions if need be. Also, this form of hiding task goals from the temporal abstractions has been shown to be effective for transfer in several multi-task hierarchical RL approaches (Dayan & Hinton, 1993; Heess et al., 2016; Jaderberg et al., 2016; Nachum et al., 2018). Separately, note that our assumption of access to task encodings is standard in many multi-task RL (Beattie et al., 2016; Plappert et al., 2018) and robotics domains (Kolve et al., 2017; Deitke et al., 2020).

We motivate the choice of allowing the manager to pick between option-policies and primitive actions as opposed to option-policies alone with the following example: consider the case where options take you to the doorways in a building with rooms. Suppose the goal was to go to the middle of some room. Now there is no policy that maps states to options that can achieve that middle-of-the-room goal. But if we allow the manager to choose both options and primitive actions, then it could get to the doorway of the room by traveling from doorway to doorway and then pick primitive actions to

get to the middle of the target room. This is a far more flexible use of options. Indeed, this is the way options were presented originally Sutton et al. (1999).

More recent works of Mann et al. (2015) and Jong et al. (2008) also argue for using primitive actions along with options to improve learning performance. Mann et al. (2015) theoretically study the role of adding primitive actions and proved that using options along with primitive actions allows a planning agent to achieve faster convergence. Jong et al. (2008) empirically studies this design choice of including primitive actions to a HRL agent and clearly show that the agent that can select between options and primitive actions can obtain optimal performance while the agent that only selects options fails. Furthermore, the work by Baumli et al. (2021) also explores HRL approaches that learn by selecting among options and primitive actions.

In the next section, we present a discovery algorithm based on meta-gradients that is used to train each network.

## 1.2 MODAC's Discovery Algorithm

**Algorithm Overview:** We extend the general meta-gradient algorithm (Xu et al., 2018) to discover options within our hierarchical architecture. It consists of two nested loops: an *inner-loop* that updates the manager to maximise the discounted sum of task rewards and that updates the option-policies to maximise the discounted sum of its corresponding option-rewards (with discounts provided by option-terminations); and an *outer-loop* that evaluates the *updated* manager and option-policies on new transitions produced by the agent and then updates the option-rewards and option-terminations by computing meta-gradients by back-propagating through the inner-loop updates.

In the inner-loop, the parameters of the manager $\theta^M$ and option-policy $\{\theta^{o_i}\}$ are updated with transitions produced by the agent on a random sample of tasks. In the outer-loop, the updated manager and option-policies are evaluated on new transitions drawn from another random sample of tasks, and then the meta-parameters of the option-reward $\{\eta^{r^{o_i}}\}$ and option-termination $\{\eta^{\beta^{o_i}}\}$ networks are updated by back-propagating through the inner-loop updates.

Below, we instantiate the algorithm for an actor-critic architecture, leaving adaptation to other RL updates to future work. A detailed derivation is presented in the Appendix.

*Inner-loop:* Consider a $n$-step trajectory $\{s_t, a_t, r_{t+1}, r^o_{t+1}, \beta^o_{t+1}, \pi^o, g\}_{t=t_0}^{t_0+n}$ generated by following an option-policy $\pi^o$ until its termination ($\beta^o_{t_0+n} = 1$), where $\pi^o$ was sampled from the manager's policy $\pi^M$ at $t = t_0$ while interacting with task $g$. For such a trajectory, the inner-loop updates to the option policy and manager parameters are:

$$\theta^o \leftarrow \theta^o + \alpha \big(G^o_t - v^o(s_t)\big) \cdot \nabla_{\theta^o} \big[ \log \pi^o(a_t|s_t) - \kappa^o v^o(s_t) \big] \tag{1}$$

$$\theta^M \leftarrow \theta^M + \alpha \big(G^M_{t_0} - v^M(s_{t_0}, g)\big) \cdot \nabla_{\theta^M} \big[ \log \pi^M(o|s_{t_0}, g) - \kappa^M v^M(s_{t_0}, g) \big] \tag{2}$$

where $\kappa^M, \kappa^o$ weight the value updates relative to the policy updates, and $G^o_t$ and $G^M_t$ are $n$-step returns for the option-policy and manager:

$$G^o_t = \sum_{j=1}^{n} (1 - \beta^o_{t+j})^j r^o_{t+j} + (1 - \beta^o_{t+n})^{n+1} v^o(s_{t+n}) \tag{3}$$

$$G^M_t = \sum_{j=1}^{n} \gamma^j r_{t+j} - \gamma^n c + \gamma^{n+1} v^M(s_{t+n}) \tag{4}$$

where $c$ is a *switching cost* added, on option terminations, to the per-step rewards used in manager's update. *The switching cost hyperparameter encourages the manager to pick options that are temporally-extended* (therefore aiding their discovery).

*Outer-loop:* In the outer-loop update to the option-reward and option-termination meta-parameters, we use a different trajectory $\{s_t, a_t, r_{t+1}, \pi^{o_t}, g\}_{t=t_0+n+1}^{t_0+n+m}$ generated by interacting with the environment using the latest inner-loop parameters $\theta^{o_i}, i = 1 \dots K$. Since this trajectory is used to evaluate the change made to the manager and option-policy parameters, we refer to it as a *validation* trajectory. The task $g$ may be different from the one used in inner-loop update. On this validation trajectory, we compute the meta-update to the option-reward and option-termination, back-propagating through the

inner-loop updates[4]:

$$\forall i, \ \eta^{r^{o_i}} \leftarrow \eta^{r^{o_i}} + \alpha_\eta \big(G_t^M - v^M(s_t, g)\big) \nabla_{\eta^{r^{o_i}}} \log \pi^{o_i}(a_t|s_t)$$

$$\forall i, \ \eta^{\beta^{o_i}} \leftarrow \eta^{\beta^{o_i}} + \alpha_\eta \big(G_t^M - v^M(s_t, g)\big) \nabla_{\eta^{\beta^{o_i}}} \log \pi^{o_i}(a_t|s_t) \tag{5}$$

*Pseudocode:* The algorithm for training a MODAC agent is summarised in Alg. 1. MODAC utilises the inner-loop updates (see Eqns.1, 2) to train the parameters of the option-policies and manager, and in the outer-loop, discovers the option-reward and termination parameters via meta-gradients obtained using the updated option-policy parameters (see Eqn. 5). *Furthermore, meta-gradients are efficiently computed through backward-mode autodifferentiation, thus making its computational complexity similar to that of the forward computation* (Griewank & Walther, 2008).

---

**Algorithm 1** Meta-gradient algorithm for option discovery

---

Initialise parameters $\theta^M$, $(\theta^{o_i}, \eta^{r^{o_i}}, \eta^{\beta^{o_i}}) \ \forall i = 1 \ldots K$
Sample task $g \sim \mathcal{G}$, state $s \sim S_0(g)$, option-policy $o \sim \pi^M(s, g)$
**repeat**
  **for** $l = 1, 2,$ **to** $L$ **do**
    # **L denotes number of inner-loop updates**
    Re-sample $(s, g, o)$ when starting a new episode
    **if** $\beta^o(s) == 1$ **then**
      $o \sim \pi^M(s, g)$
    **end if**
    Obtain transition using option-policy $s \sim E(s, \pi^o(s))$      # **E denotes the environment**
    # **Inner-loop update**
    Update option-policy $\theta^o$ with Eqn. 1
    Update manager $\theta^M$ on states where it samples $o$ with Eqn. 2
  **end for**
  Obtain transitions from another task using the updated manager $\theta^M$ and option-policy $\{\theta^{o_i}\}$ parameters
  # **Outer-loop update**
  Update option-reward $\eta^{r^{o_i}}$ and termination $\eta^{\beta^{o_i}}$ with Eqn. 5    $\forall i = 1 \ldots K$
**until** maximum number of timesteps

---

### 1.3 Differences Between MODAC and Prior Option-Discovery Methods

Previous option-discovery works that have been explored in a multi-task setup, (c.f. MLSH, Option-Critic) usually train all the options to optimise the task reward for the current task generating the data, though gradients are accumulated across tasks. In contrast, in our agent, each option-policy optimises a *different* objective, parameterised by the corresponding task-independent option-reward and termination that are discovered to be *directly* useful across many tasks. The MODAC architecture and the meta-gradient learning updates discover task-independent, general-purpose, disentangled options that can help the agent not just to achieve higher rewards during training, but also to speed up learning in new tasks.

## 2 Empirical Results

We empirically evaluated MODAC using a mix of gridworld and navigation tasks from DeepMind Lab. Our main numeric metric for the quality of the options discovered by our agent was their usefulness for transfer. In addition to this, we evaluated their quality in various ways, including whether they were temporally extended and whether they are diverse in their behaviour. Our experiments all included a *training* phase followed by a *testing* phase.

*Training:* For every episode, a training task was selected randomly from a set of training tasks that were distinct from the test tasks. In this phase, MODAC jointly learned the manager, option-policies, option-rewards, and option-terminations with transitions from the episode. As noted earlier, in this

---

[4]The option-policy parameters are a function of option-reward and termination parameters, and thus, allow for computing the meta-gradients.

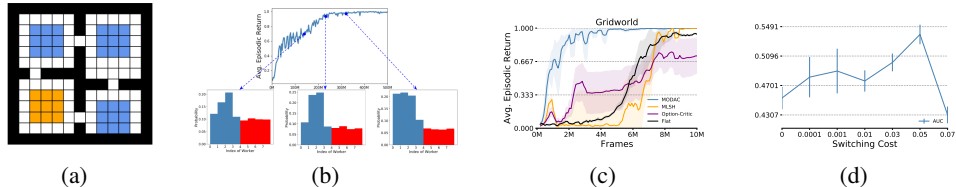

|  (a)  |  (b)  |  (c)  |  (d)  |

Figure 2: (a) shows the gridworld's layout, the subset of goal locations (in light blue) that are used as training tasks and the disjoint set of goal locations (in orange) used as test tasks. (b) shows how MODAC chose among options and primitive actions at three different points during training, as those options were being discovered. In the histogram, each of blue and red bars measure the frequency of selecting an option and primitive action respectively. From this, we can see that the agent picked options more often than primitive actions throughout the training phase. (c) shows the (transfer) performance on held-out tasks of options learned by MODAC, MLSH, Option-Critic, and a Flat agent. The agent with options discovered by MODAC learned significantly faster than other baselines, demonstrating the usefulness of those options during transfer. (d) shows MODAC's average (transfer) performance, as a function of switching cost $c$ used during training.

phase a penalty in the form of switching cost (a negative reward) is added to the task rewards that is used in the manager's learning update (see Eqn. 2, 4). It is based on the intuition of incentivising the manager to make fewer decisions while trying to maximize rewards from a given task. It does this by discouraging the manager from selecting primitive actions too often as it incurs a lot of switching costs, thereby decreasing its return. The switching cost is used with every update to the manager during the training phase and the role of switching cost is to encourage the manager to pick and thus help discover temporally-extended options by penalising the selection of primitive actions too often.

*Testing:* We froze option-policies and option-termination functions after training, and transferred them to a new manager with randomly initialised parameters; then evaluated how fast the manager could learn on different unseen test tasks. Performance on each test task was evaluated independently, re-initialising the manager every time, and the results presented are averages across test tasks. If the manager finds the transferred options to be useful for maximising rewards from test tasks, then the manager would naturally pick those options without need of the switching cost penalty. Thus the switching cost is not used in this phase.

*Baselines:* We compared the performance of MODAC at test time with that of *Flat*, a non-hierarchical actor-critic agent, and also to the performance of hierarchical agents that used options discovered using MLSH and a multi-task extension of the Option-Critic, as these hierarchical approaches also use multi-task performance on rewards to drive their discovery process. We use an identical train/test setup for all the hierarchical agents. The hierarchical approaches learn to select among the union of the set of option-policies and the set of primitive actions, which is identical to MODAC. Details on all the agents, their hyperparameter choices and other implementation details are in the Appendix.

## 2.1  An Illustrative Gridworld

**Domain Description:** Consider a simple gridworld with 4 connected rooms, whose layout is shown in Fig. 2a. The agent receives a reward of 1 on reaching the goal and 0 at every other time step. During training, at the start of each episode, a new training-goal location is randomly chosen from the set shown in blue in Fig. 2a. During testing, a goal position is chosen randomly from a disjoint set shown in orange in Fig. 2a and remains fixed for all episodes. The agent's observations included its current location and the grid's layout (in 2 separate image channels). The agent was also fed the goal location during training (as the 3rd channel). Each test task was evaluated separately and the overall transfer performance was computed by averaging across all test runs.

**Visualisation:** Fig. 3 shows the option-policies discovered in the gridworld (arrows indicate the direction of movement), at the end of training. We found the discovered options to make intuitive sense, given the distribution of training tasks. The first three options each led into one of the rooms where the training goal states (in blue) were concentrated: specifically, from any state the options shown in Fig. 3a−c steered the agent to the upper-left, upper-right and lower-right rooms, respectively. The option in Fig. 3d seems redundant given the options in Fig. 3b and c. We hypothesise that such

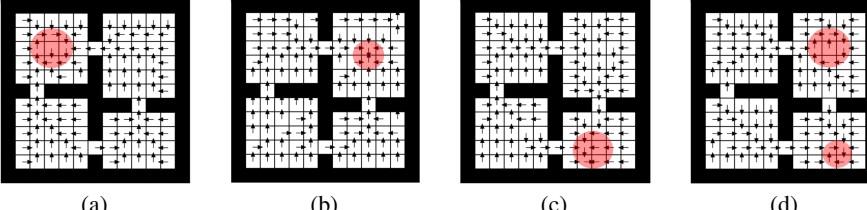

Figure 3: The red circle approximately marks the destination/subgoal states for each discovered option-policy (arrows indicate the direction of movement). Options (a−c) each led the agent directly to one of the three areas where the training tasks were concentrated. Option (d) appeared to be redundant and led to either one of two such areas depending on the start state.

redundancy was due to the training task distribution only featuring goals in three rooms, thus requiring only three options to capture most of the structure in the space of behaviours required by the training tasks.

**Quantitative Analysis:** (1) We measured how the manager selected among options and primitive actions at three points during *training*. Fig. 2b plots the manager's choices as histograms. The 4 blue bars denote how often each of the 4 options was picked, red bars depict how often primitive actions were chosen. The manager continued to pick both options and primitive actions throughout training. Consistent with the option-redundancy identified in Sec. 2.1, only 3 of the 4 options were selected frequently towards the end of training.
(2) Option-policies lasted 5.46 steps on average, and so were temporally extended and given how often options are picked were overwhelmingly responsible for behaviour.
(3) Fig. 2c shows the performance on *test tasks*, when a randomly initialised manager was provided with the (fixed) options discovered by MODAC at the end of the training phase from 6 independent runs (10M frames were used for training). The speed of learning on the test tasks was substantially faster compared to a Flat agent that only used primitive actions; it also outperformed the baselines with access to the options discovered by MLSH or the Option-Critic.
(4) During transfer to test tasks, the manager selected options 56.11% of the time. But recall that options last more than 5 steps and so in effect options controlled behavior more than 85% of the time.
(5) Fig. 2d shows transfer performance (averaged throughout the test phase) as a function of the switching cost used in training. We found a sweet spot at a cost of 0.05, for which the discovered options were maximally useful.

## 2.2 Scaling Up to DeepMind Lab

We applied MODAC to DeepMind Lab (Beattie et al., 2016), a challenging suite of RL tasks with consistent physics and action spaces, and a first-person view as observations to the agent (hence there is partial observability). We evaluated our approach on 4 different task **sets** (see titles of Fig. 6 for their names), where each set corresponds to a different type of navigation problem. For instance, the task set `explore_goal_locations` requires the agent to explore a maze to visually identify and then reach a goal location identifiable by a special 3D marker. Each set includes 'simple' and 'hard' variants; in simple variants the layout includes at most a handful of rooms, while harder variants require navigation in large mazes. In both cases, procedural generation is used to create a different layout for every episode. The challenge is to discover options on simple tasks that can be useful to learn faster on hard tasks. More details in Appendix.

**Visualisation:** In `explore_goal_locations`, the state space is obviously too large to visualise entire option policies in one plot as we did for the gridworld. Instead, we generated sample trajectories of experience, and used the DEBUG information provided by DeepMind Lab environments (which is not fed to the agent) to visualise trajectories by drawing the path taken by the agent onto a top-down view of the maze. The segments corresponding to primitive actions were coloured in blue, while those corresponding to the discovered options were assigned an arbitrary different colour. The agent's start and end locations were highlighted by white and green circles, respectively.

We generated sample trajectories using a trained manager with access to both discovered options and primitive actions. When an option is selected, it is executed until the option-termination function is

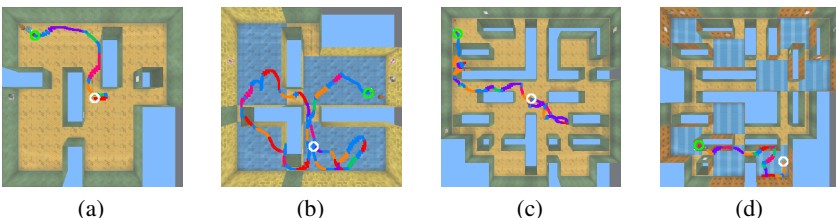

|     |     |     |     |
| :-: | :-: | :-: | :-: |
| (a) | (b) | (c) | (d) |

Figure 4: Option Execution by a Manager. The figures show 4 distinct samples of trajectories generated by a manager with access to both discovered options (marked in non-blue colours) and primitive actions (in blue). (a) and (b) correspond to mazes in the training set. (c) and (d) are sampled from the testing set. The agent's starting and final positions are highlighted by a white and green circles, respectively. In all cases, the agent successfully achieves the task's objective of reaching the rewarding goal location by using a mixture of primitive actions and discovered options.



|     |     |     |     |
| :-: | :-: | :-: | :-: |
| (a) | (b) | (c) | (d) |

Figure 5: Sampled Option-Policies. The task is to reach the rewarding goal location that was always in the top left corner but not in the line of sight for the agent at the start of the episode. The agent's starting and option-termination positions are highlighted by a white and green circles. In (a) and (b) we show the (very different) trajectories followed by two different option-policies when initialised in the same state within a maze from the training set. In (c) and (d) we show two trajectories in a maze from the test set, for the same pair of option-policies (again initialised in the same state).

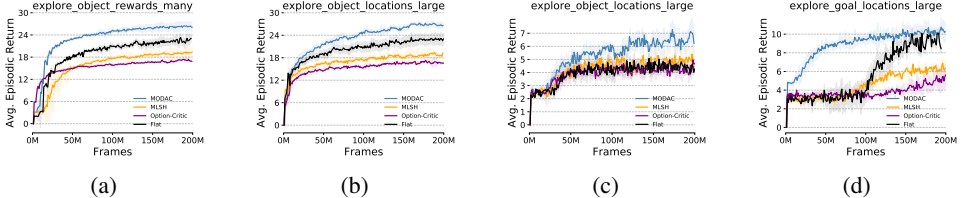

|     |     |     |     |
| :-: | :-: | :-: | :-: |
| (a) | (b) | (c) | (d) |

Figure 6: Transfer experiments on DeepMind Lab. Options, discovered from training tasks, were transferred to the corresponding test tasks for hierarchical agents. Figures show the (transfer) learning performance of different agents while they learned to maximise rewards in the test task. MODAC with discovered options learned better and thus was able to achieve better asymptotic performance on 3 of those 4 new, unseen tasks, while learning substantially faster on the 4th task.

triggered, at which point the manager chooses again. The manager made extensive use of the learned options in both the training tasks, for which a sample is shown in Fig. 4a and b, and in the testing tasks, shown in Fig. 4c and d. In the simple training tasks, the manager often only required a handful of option executions to reach the goal. For instance, in Fig. 4a, after the goal (top-left corner) enters in the line of sight of the agent, it only takes two options to take the agent to the goal. In the hard test task, the manager still relied extensively on the learned options, but it needed to chain a higher number of them during the course of an episode. In Fig. 4c, the agent was spawned in the centre of the maze at the beginning of an episode. The agent used a mix of options and primitive actions in order to identify and reach the goal location (top-left). In both the training and test tasks, we often found that the manager used options extensively to explore the maze, but it relied on primitive actions for the *last mile navigation* (i.e. in proximity of the goal).

Fig. 5 visualised trajectories generated by following 2 discovered option-policies on the training task set. After seeding the episode in the same way, we observed that the option-policies produced quite diverse behaviours. For instance, we found that the options in Fig. 5a and b explored the maze in very different ways. We note that when the latter entered in line of sight of the goal – in the top left corner – it marched straight into it. In Fig. 5c and d, we visualised the execution of the same pair of options when triggered on the larger maze from the testing task set; while neither happened to encounter the goal marker in those two episodes, both demonstrated a meaningful temporally extended behaviour that resulted in good exploration of a vast portion of the maze.

**Quantitative Analysis:** In each of the 4 training task-sets we discovered 5 options (using a switching cost $c = 0.03$), and observed that the average length of the options was 12 steps. Fig. 6 shows the performance on testing task sets, averaged across 6 independent runs, for randomly initialised managers, given access to the pre-trained options discovered by MODAC (200M frames were used for training). Those agents learned to maximise rewards faster than the Flat agent that learned with primitive actions alone, and reached higher asymptotic performance in 3 of them. The transfer performance with MODAC-discovered options was also better than that of the MLSH and Option-Critic baselines in all 4 domains. We again measured the distribution of the manager's choices at transfer time: options were selected 63.76% of the time, which given that options last about 12 steps implies that our discovered options were responsible for behaviour more than 95% of the time.

**Additional Results:** We studied the effect of MODAC's transfer performance when fewer samples are used in the training phase. Fig. 7 shows performance on one DeepMind Lab task set when only 5M and 10M samples are used; *in this case the x-axis includes the training samples for the MODAC curves*. The MODAC agent learned faster than the Flat agent, showing that useful options can be discovered with small enough training samples to outperform the Flat agent in a comparison that takes all samples into account. In Appendix, we present results of additional studies carried out to answer the following empirical questions: (1) How are the options qualitatively different when the number of options (a hyperparameter) is varied? (2) Could MODAC discover options in Atari games from unsupervised training tasks that could be useful to maximise the game score at test time? In addition we provide more qualitative visualisations from the MODAC agent and its option-policies discovered from DeepMind Lab task sets.

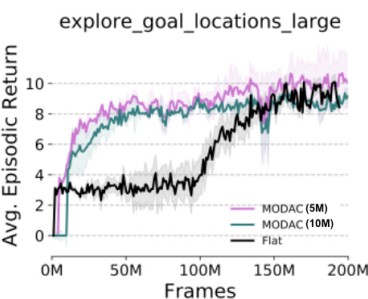

Figure 7: Shows (transfer) performance on a DeepMind Lab task set when fewer samples (5M and 10M) are used for discovering options during training, compared to 200M used in our main experiments. Learning curves for MODAC are right-shifted to account for the number of samples used in training.

## 3   Related Work

**Hierarchical Reinforcement Learning (HRL):** Prior HRL work has shown that temporal abstractions lead to faster learning in both single-task (Sutton et al., 1999; Dayan & Hinton, 1993; Dietterich, 2000; Ghavamzadeh & Mahadevan, 2003; Kulkarni et al., 2016) and multi-task/transfer setup (Konidaris & Barto, 2007; Brunskill & Li, 2014; Tessler et al., 2017). The aforementioned works require hand-designed temporal abstractions; which can be challenging in scenarios where the agent interacts with a distribution of tasks. In contrast, we aim to discover temporal abstractions purely from experience *without* human supervision or domain knowledge.

**Discovery of Temporal Abstractions:** Majority of the prior HRL work for discovering abstractions are either *unsupervised* approaches[5] or operate in a single-task setup. Older unsupervised approaches to discovery have exploited graph-theoretic properties of the environments such as bottlenecks (McGovern & Barto, 2001), centrality (Şimşek et al., 2005), and through clustering of states (Mannor et al., 2004). More recent unsupervised approaches discover options by learning proto-value functions (Machado et al., 2017), successor feature representations (Machado et al.,

---

[5]By unsupervised, we refer to approaches that do not require extrinsic reward signal to drive the discovery process.

2018), entropy minimisation (Harutyunyan et al., 2019), maximising empowerment (Gregor et al., 2016), maximising diversity (Eysenbach et al., 2018) or through probabilistic inference (Ranchod et al., 2015; Daniel et al., 2016). Recent approaches for discovery in single-task setup learn agents with architectures that are inspired from options framework (Vezhnevets et al., 2016; Bacon et al., 2017; Kostas et al., 2019) and feudal RL framework (Jaderberg et al., 2016). In contrast to the aforementioned approaches, our approach is motivated by the multi-task RL setup where the objective is to discover task-independent options that can be reused by the RL agent to master all of its tasks.

Among prior work that investigates discovery in *multi-task* settings, Meta-Learning Shared Hierarchies (MLSH) learns a set of option-policies by directly optimising task rewards via a joint actor-critic update. In contrast, our approach discovers options through the subgoals flexibly defined by option-rewards and terminations; also, our approach learns the temporal scale of each option, while most prior approaches, including MLSH, fix the temporal scale.

## 4   Conclusions

We introduced MODAC, a novel hierarchical agent and a meta-gradient algorithm, for option discovery in a multi-task RL setup. Through visualisations from gridworld and DeepMind Lab, we showed that the options discovered capture diverse and important properties of the behaviours required by the training task distribution. Through DeepMind Lab, we show that MODAC is scalable, and thus, viable for option discovery in challenging RL domains. A promising direction for future work is to investigate whether meta-gradients can also be applied to the discovery of options suitable for planning in *model-based* RL.

## Acknowledgement

Part of this work was conducted at the University of Michigan by Vivek Veeriah where he is supported by DARPA's L2M program. Any opinions, findings, conclusions, or recommendations expressed here are those of the authors and do not necessarily reflect the views of the sponsors.

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
