# Supplementary Material:
# Discovery of Options via Meta-Learned Subgoals

## A  Potential negative societal impact

While all AI advances can have potential negative impact on society through their misuse, this work advances our understanding of fundamental questions of interest to RL and at least at this point is far away from potential misuse.

## B  Additional Qualitative Experiments on Gridworld

For the main gridworld results in text, we discovered $4$ options ($K = 4$, where $K$ is the number of options; a hyperparameter) and visualised them. This additionally brings a question which is what do these options look like when this hyperparameter is set to a different value. In this subsection, we provide visualisations for the options discovered when MODAC is trained with $K = 2$ (see Fig. 1) and $K = 8$ (see Fig. 2).

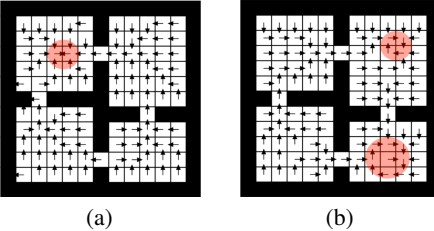

(a)                      (b)

Figure 1: Option visualisations on the four-room gridworld when 2 options were discovered. The red circle approximately marks the destination/subgoal states for each discovered option-policy. Option (a) led the agent to the upper-left room, whereas option (b) led to either the upper-right or lower-right rooms depending on the start state.

### B.1  Additional Qualitative Visualisations from DeepMind Lab

In addition to the visualisations in the main text, we include here additional visualisations of trajectories obtained by executing all the discovered option-policies on a training task (Fig. 3) and on its corresponding test task (Fig. 4). From these visualisations, it can be observed that each of the option policy do produce diverse and structured exploratory behaviours in both training and test tasks.

We also include visualisations of a trained MODAC agent picking options to produce behaviour in order to complete an episode on a training task (Fig. 5) and in its corresponding test task (Fig. 6). In these Figures, primitive actions are coloured in blue and options are coloured in an arbitrary, different colour. The agent's start state and end state are highlighted with white and green circles respectively.

35th Conference on Neural Information Processing Systems (NeurIPS 2021).

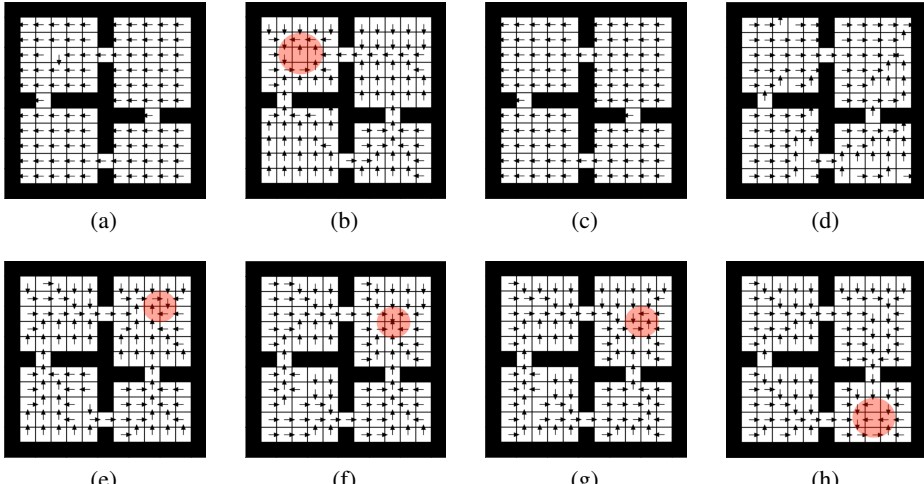

Figure 2: Option visualisations on the four-room gridworld when 8 options were discovered. The red circle approximately marks the destination/subgoal states for each discovered option-policy. Option (b, e, f, g, h) each led the agent to one of the rooms where the training tasks are concentrated. The options (a, c) were similar and seem to move the agent in the left cardinal direction. Option (d) doesn't seem to have a well-defined subgoal. We hypothesise that the options (b, e, f, g, h) were picked often compared to the options (a, c, d), during training time, which led them to have well-defined subgoals.

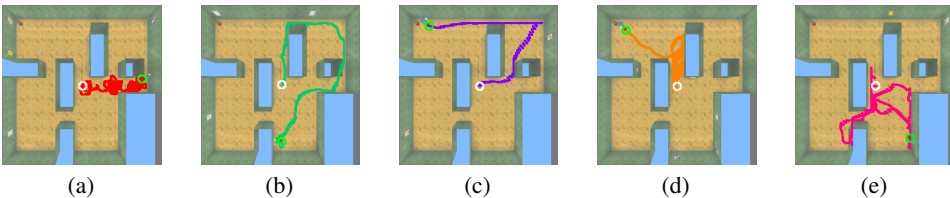

Figure 3: Sampled Option-Policies on a Training Task. The goal was always in the top-left corner but not in the line of sight for the agent at the start of the episode. The agent's starting and end positions are highlighted by white and green circles. Each figure shows a trajectory by following each of the 5 discovered options.



Figure 4: Sampled Option-Policies on a Test Task. The goal was always in the top-left corner but not in the line of sight for the agent at the start of the episode. The agent's starting and end positions are highlighted by white and green circles. Each figure shows a trajectory by following each of the 5 discovered options (which are obtained from the training phase).

These figures show that the MODAC agent relies on picking options for producing behaviour in both training and test task, thus validating that our approach does indeed learn reusable and transferrable options, which is the primary reason behind their improved transfer performance.

## B.2 Additional Experiments on Atari

Here, we study the question of whether MODAC can discover options in Atari games from unsupervised learning tasks that could become useful to maximise the game score at test time.

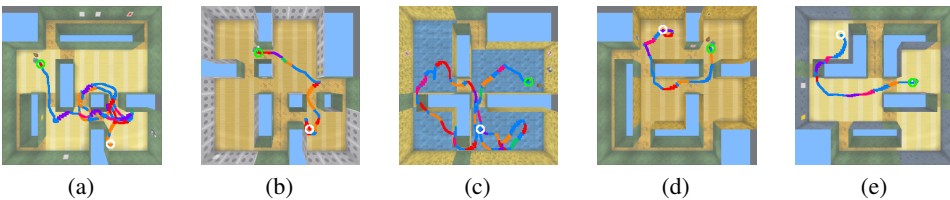

Figure 5: Option Execution by a Manager on a Training Task. The figures show 5 distinct samples of trajectories generated by a manager with access to both discovered options (marked in arbitrary colours) and primitive actions (in blue). The agent's starting and final positions are highlighted by a white and green circles, respectively. In all cases, the agent successfully reaches the goal by using a mixture of primitive actions and discovered options.



Figure 6: Option Execution by a Manager on a Test Task. The figures show 5 distinct samples of trajectories generated by a manager with access to both discovered options (obtained from the training phase; marked in arbitrary colours) and primitive actions (in blue). The agent's starting and final positions are highlighted by a white and green circles, respectively. In all cases, the agent successfully reaches the goal by using a mixture of primitive actions and discovered options.

Atari games, unlike DeepMind Lab tasks, have mutually inconsistent game dynamics and thus the problem of discovering options useful across distinct games would require significant new work on a separate problem, that of learning cross-game abstractions that can then support shared options. Therefore, we considered each Atari game as a separate test domain and, instead, procedurally generated multiple training tasks within each game. Specifically, we used pixel-control tasks, defined by Jaderberg et al. (2016), as our set of generated unsupervised training tasks. Those tasks were quite different from the test task, which was the usual task of maximising the Atari game score. Importantly, in defining the training tasks, we ignored episode terminations in the pixel-control task definition to avoid any information leaks from the test task. The challenge for MODAC was to use the generated pixel-control tasks to discover options that could speed up learning if provided to a randomly initialised manager solving the corresponding Atari game. Note that this is quite different from the typical use of pixel-control tasks, where they are used to aid learning of good state representations. In our case, the manager policy did not share any weights with the option-policy and termination networks, therefore any improvements in learning efficiency can only be attributed to the options themselves, and not to representation learning.

**Quantitative Analysis:** We choose 4 Atari games (*Boxing, Hero, MsPacman, Riverraid*), where pixel-control was separately found useful for representation learning. We discovered 5 options (with a switching cost $c = 0.1$), and their average length was 7 steps. Fig. 7 shows the transfer performance of a randomly initialised manager, when given access to the pre-trained options discovered by MODAC on the pixel-control tasks defined on the corresponding Atari game. In all 4 games, the agent that had access to the options discovered by MODAC learned to maximise game-score rewards much faster than a Flat agent learned using primitive actions alone. We measured the distribution of the manager's choices at transfer time and observed that options were selected $60.79\%$ of the time, which implies that our discovered options were responsible for agent's performance. In MsPacman, the Flat agent learned faster but saturated at a lower level, perhaps showing that the use of options during transfer can help explore better. This is consistent with the findings by Tessler et al. (2017) that options help initially for exploration. In all 4 games, the transfer-learning performance of MODAC is much better than the transfer-learning performance of MLSH and Option-Critic.

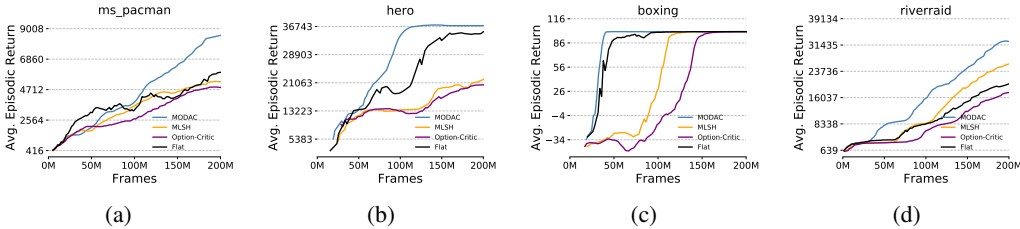

(a)                          (b)                          (c)                          (d)

Figure 7: Transfer experiments on Atari games. Figures show the performance of different agents learning to maximise rewards from the main task while having access to options discovered from pixel-control tasks defined on that same game. MODAC with discovered options learned faster in all 4 Atari games and thereby was able to achieve better asymptotic performance on 3 of these 4 Atari games.

Note that during the training phase, options were discovered from the pixel-control training tasks by ignoring the episode terminations (i.e., unsupervised). This was done deliberately in order to avoid leaking of any task-relevant information from the test task (which is to maximise the Atari game score). Here, we look at the effect of discovering options when episode terminations are not ignored.

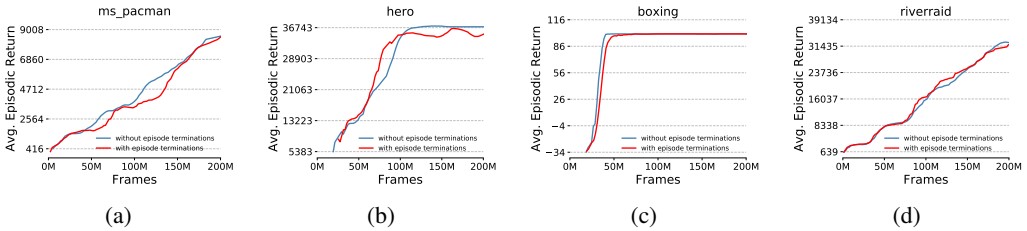

(a)                          (b)                          (c)                          (d)

Figure 8: Transfer experiments on Atari games. Figures show the performance of MODAC agents learning to maximise rewards from the main task while having access to options discovered from pixel-control tasks defined on the same game. The figure compares MODAC agents whose options were discovered by ignoring (in blue) and including (in red) the episode termination.

In Figure 8, we present learning curves of the MODAC agent with options discovered with and without episode terminations from the training task. From these, we can see that MODAC agent with options discovered by ignoring the episode terminations learns a little faster than its counterpart that included episode terminations in 2 out of 4 games (namely, *MsPacman, Boxing*). In *Hero*, ignoring the episode terminations seems to allow MODAC to achieve a stable asymptotic performance, and in *Riverraid*, there does not seem to be any visible difference in performance.

## B.3 Objective and Update Equations for Discovering Option-Rewards and Terminations Using Meta-gradients

The objective of MODAC is to discover the parameters of the option-rewards $\{\eta^{r^{o_i}}\}$ and terminations $\{\eta^{\beta^{o_i}}\}$ so as to maximise the hierarchical agent's performance $G_t^M$ through the parameters of the option-policies $\{\theta^{o_i}\}$ that they induce. One way of accomplishing this objective is by measuring a change in the option-rewards and terminations on the hierarchical agent's performance through the change they induce in the option-policies.

Recall that the parameters of option-policies are learned to maximise their corresponding option-rewards, with discounts applied with corresponding option-terminations, on the local trajectories that they produced; while the parameters of the manager's policy $\theta^M$ are learned to maximise extrinsic rewards (see Eqns. 1, 2 from main text).

The objective for option-rewards and option-terminations is to maximise the following:

$$\max_{\{\eta^{r^{o_i}}\},\{\eta^{\beta^{o_i}}\}} \mathbb{E}_{\{\theta^{o_i}\},\theta^M,\mathcal{G}}\left[G_t^M\right]$$

where the $n$-step return for the manager is defined as: $G_t^M = \sum_{j=1}^n \gamma^j r_{t+j} - \gamma^n c + \gamma^{n+1} v^M(s_{t+n})$, and $c$ is the switching cost. The expectation of the agent's performance is over the parameters of option-policies, manager's policy and set of training tasks $\mathcal{G}$.

Using the score-function estimator (similar to its use in the policy-gradient theorem (Sutton et al., 2000)) and chain-rule, we can obtain the update equation for the option-rewards and terminations as follows:

$$\forall i, \eta^{r^{o_i}}{}' = \eta^{r^{o_i}} + \alpha_\eta \nabla_{\eta^{r^{o_i}}} \mathbb{E}_{\{\theta^{o_i}\}, \theta^M, \mathcal{G}}\left[G_t^M\right]$$

$$\approx \eta^{r^{o_i}} + \alpha_\eta \mathbb{E}_{\{\theta^{o_i}\}, \theta^M, \mathcal{G}}\left[G_t^M . \nabla_{\theta^{o_i}{}'} \log \pi^{o_i} . \nabla_{\eta^{r^{o_i}}} \theta^{o_i}{}'\right]$$

$$\forall i, \eta^{\beta^{o_i}}{}' = \eta^{\beta^{o_i}} + \alpha_\eta \nabla_{\eta^{\beta^{o_i}}} \mathbb{E}_{\{\theta^{o_i}\}, \theta^M, \mathcal{G}}\left[G_t^M\right]$$

$$\approx \eta^{\beta^{o_i}} + \alpha_\eta \mathbb{E}_{\{\theta^{o_i}\}, \theta^M, \mathcal{G}}\left[G_t^M . \nabla_{\theta^{o_i}{}'} \log \pi^{o_i} . \nabla_{\eta^{\beta^{o_i}}} \theta^{o_i}{}'\right]$$

where $\theta^{o_i}{}'$ refers to the inner-loop option-policy parameters obtained by making an inner-loop update, given produced by Eqn. 1 (thus, they are differentiable w.r.to option-rewards and terminations).

The gradients for parameters of the option-rewards $\eta^{r^{o_i}}$ and terminations $\eta^{\beta^{o_i}}$ are obtained by following a policy-gradient update to maximize extrinsic returns $G^M$. The policy for the learning agent can be viewed to be a factored policy which is composed of the manager's policy which selects an option and this selected policy then selects an action which is executed in the environment. The meta-gradient is then computed by differentiating through this factored policy where the parameters of the manager and option-policies are obtained after (atleast) an inner-loop update (In our work, we performed 5 inner-loop updates). The gradient term for the option-rewards and terminations (i.e meta-parameters) are approximate because it ignores the effect of a change in option-rewards and terminations on the manager's policy. The update term only captures the direct dependence of the meta-parameters on the option-policies.

$$\eta^{r^{o_i}}{}' = \eta^{r^{o_i}} + \alpha_\eta \nabla_{\eta^{r^{o_i}}} \mathbb{E}_{\theta^{o_i}, \theta^M, \mathcal{G}}\left[G^M\right]$$

$$= \eta^{r^{o_i}} + \alpha\eta \mathbb{E}\theta^{o_i}, \theta^M, \mathcal{G}\left[G^M \nabla\eta^{r^{o_i}} \log \left[\pi^M(o_i|s,g)\pi^{o_i}(a|s)\right]\right]$$

$$\approx \eta^{r^{o_i}} + \alpha_\eta \mathbb{E}\theta^{o_i}, \theta^M, \mathcal{G}\left[G^M \nabla\eta^{r^{o_i}} \log \pi^{o_i}(a|s)\right]$$

$$\eta^{r^{o_i}}{}' \approx \eta^{r^{o_i}} + \alpha_\eta \mathbb{E}\theta^{o_i}, \theta^M, \mathcal{G}\left[G^M \nabla\theta^{o_i}{}' \log \pi^{o_i}(a|s) \nabla_{\eta^{r^{o_i}}} \theta^{o_i}{}'\right]$$

$$\eta^{\beta^{o_i}}{}' = \eta^{\beta^{o_i}} + \alpha_\eta \nabla_{\eta^{\beta^{o_i}}} \mathbb{E}_{\theta^{o_i}, \theta^M, \mathcal{G}}\left[G^M\right]$$

$$= \eta^{\beta^{o_i}} + \alpha\eta \mathbb{E}\theta^{o_i}, \theta^M, \mathcal{G}\left[G^M \nabla\eta^{\beta^{o_i}} \log \left[\pi^M(o_i|s,g)\pi^{o_i}(a|s)\right]\right]$$

$$\approx \eta^{\beta^{o_i}} + \alpha_\eta \mathbb{E}\theta^{o_i}, \theta^M, \mathcal{G}\left[G^M \nabla\eta^{\beta^{o_i}} \log \pi^{o_i}(a|s)\right]$$

$$\eta^{\beta^{o_i}}{}' \approx \eta^{\beta^{o_i}} + \alpha_\eta \mathbb{E}\theta^{o_i}, \theta^M, \mathcal{G}\left[G^M \nabla\theta^{o_i}{}' \log \pi^{o_i}(a|s) \nabla_{\eta^{\beta^{o_i}}} \theta^{o_i}{}'\right]$$

The update equations are a stochastic gradient update and are computed over samples obtained from the environment. They are used according to how they are described in Alg. 1, where multiple inner-loop updates are performed per outer-loop update to option-rewards an terminations. Since the gradients for option-rewards and terminations are computed through the parameters of the option-policies, we call them meta-gradients.

## B.4   Neural Network Architecture

The MODAC agent, which consists of a manager, option-policy, option-reward and option-termination network, uses an identical torso architecture for all of them; and details about the torso are described below.

*Gridworld:* The neural net torso consisted of a 2-layer CNN each with 32 filters (filter size $= 2 \times 2$, with stride length $= 1$). The activations from the CNN were transformed by a single fully-connected layer of size 256.

*DeepMind Lab:* We use a Deep ResNet torso identical to the one from Espeholt et al. (2018), with an additional LSTM layer (with 256 hidden units) after the feed-forward torso.

*Atari:* We use a Deep ResNet torso identical to the one from Espeholt et al. (2018).

All layers of the neural network use a ReLU activation function in the intermediate layers. The output layers of the option-reward and option-termination function use a $arctan$ and $sigmoid$ activations respectively.

For the training phase in the gridworld, the task information was added as an additional channel to the input image, which was given as input to the manager network. In DeepMind Lab and Atari, during training, we obtain the task information as a one-hot vector from the environment and is passed through an embedding network to produce a 128-dimensional vector. This vector is concatenated with the feed-forward produced by the Deep ResNet torso of the manager network, which is then used for subsequent computations to produce the manager's policy.

We also used identical architecture choices for the hierarchical baselines.

### B.5 Preprocessing

For both DeepMind Lab and Atari domains, the input to the learning agent consists of 4 consecutively stacked frames where each frame is a result of repeating the previous action for 4 time-steps, greyscaling and downsampling the resulting frames to $84 \times 84$ images, and max-pooling the last 2. These are fairly canonical preprocessing pipeline applied to DeepMind Lab and Atari environments, and additionally, rewards are clipped to the [-1, 1] range.

### B.6 Hyperparameters

For both hierarchical and flat agents (MODAC, Option-Critic, MLSH, actor-critic), we tuned the following hyperparameters: entropy weight and learning rate. In the case of the hierarchical agents, we tied the entropy weights for manager and option-policies to take the same value. Similarly, we also tied the learning rates used for training the parameters of the manager and option-policies. For MODAC, we used a single learning rate for learning the parameters of option-reward and option-termination.

The hierarchical agents also include a switching cost hyperparameter, which is separately tuned for each agent.

We tuned the hyperparameters for each agent separately and then used a single set of hyperparameters across all DeepMind Lab and Atari environments, for each agent (which is usually the norm in many Deep RL work). The hyperparameters that we found for MODAC after tuning are reported in Table 1.

We considered the following set of values $\{0.0001, 0.001, 0.01, 0.03\}$ for tuning the entropy weights and correspondingly $\{0.0001, 0.0003, 0.0006, 0.001, 0.003\}$ for the learning rates. For switching cost, we searched over $\{0, 0.0001, 0.001, 0.01, 0.03, 0.05, 0.07, 0.1\}$. Furthermore, we used RM-SProp as the optimiser for updating the parameters of the learning agents.

### B.7 Experimental Setup for DeepMind Lab

For our experiments on DeepMind Lab, we evaluated our approach on 4 different task sets, where each set corresponds to a different navigation problem. Each set consists of a training task and a test task; In all our task sets, the training task is simpler to learn for an actor-critic agent when compared to the test task. Below, we provide the names of the tasks from the four task sets and these are taken from the suite of DeepMind Lab tasks (Beattie et al., 2016).

| Set No. | Training Task | Test Task |
|---|---|---|
| 1 | *explore_goal_locations_small* | *explore_goal_locations_large* |
| 2 | *explore_object_rewards_few* | *explore_object_rewards_many* |
| 3 | *explore_object_locations_small* | *explore_object_locations_large* |
| 4 | *explore_obstructed_goals_small* | *explore_obstructed_goals_large* |

In both these the training and test tasks, the layout of the maze is procedurally generated, for every episode. Furthermore, the agent's start state is randomly initialised; the goal locations (for Set 1, 4), number of objects (for Set 2) and object locations (for Set 3) are also procedurally generated.

### B.8 Baselines

We compare MODAC with the following three baseline agents in all our experiments. The first two of them are hierarchical agents that discover options/skills using their respective approaches, while the third is a non-hierarchical flat actor-critic agent. Note that the Hierarchical RL baselines (i.e., MLSH and Option-Critic) also learn to select among the union of options and primitive actions, identical to our MODAC agent. Hierarchical RL baselines also use an architecture identical to that of our MODAC agent.

*Meta-Learned Shared Hierarchies (MLSH)* (Frans et al., 2017): The manager and option-policies are independently trained using an actor-critic update. The manager learns its policy by maximising task rewards; workers learn option-policies by maximising task rewards on the local trajectories generated whenever they were picked. The time scale of the workers is a fixed hyper-parameter. We tuned this via a search, and it is set to 5 in gridworld experiments and to 10 in Atari and DeepMind Lab.

*Multi-task extension of the Option-Critic with Deliberation Cost* (Harb et al., 2018): The original Option-Critic with deliberation cost was designed for a single-task setting. It uses a manager and a set of workers, which learn their policies by optimising task rewards. The workers also learn a termination through the task-value function. We extend this to our multi-task setting, mirroring the architectural choices of our agent: the manager learns a task-conditional policy, while the workers learn task-independent policies and terminations.

*Non-Hierarchical Actor-Critic (Flat):* In addition to the two hierarchical baselines described above, we also compare against a vanilla, non-hierarchical, actor-critic agent.

### B.9 Resource Usage

The average running time for each agent on the DeepMind Lab training tasks is reported below. For the hierarchical agents, the running time that during the training phase are reported is significantly higher than that of the flat actor-critic agent, as they are simultaneously learning to solve the training tasks *and* discover options. In the test phase, the hierarchical agents reuse their discovered options, and as a result, their running times are similar to that of the flat actor-critic agent.

| Agent | Running Time |
|---|---|
| Actor-Critic | 3 hours 10 mins |
| MLSH | 4 hours 31 mins |
| Option-Critic | 4 hours 46 mins |
| MODAC | 5 hours 56 mins |

### B.10 Computing Infrastructure

We run our experiments using a distributed infrastructure implemented in JAX (Bradbury et al., 2018). The computing infrastructure is based on an actor-learner decomposition (Espeholt et al., 2018), where multiple actors generate experience in parallel, and this experience is channelled into a learner via a small queue. Both the actors and learners are co-located on a single machine, where the host is equipped with 56 CPU cores and connected to 8 TPU cores (Jouppi et al., 2017). To minimise the effect of Python's Global Interpreter Lock, each actor-thread interacts with a *batched environment*; this is exposed to Python as a single special environment that takes a batch of actions and returns a batch of observations, but that behind the scenes steps each environment in the batch in C++. The actor threads share 2 of the 8 TPU cores (to perform inference on the network), and send batches of fixed size trajectories of length T to a queue. The learner threads takes these batches of trajectories and splits them across the remaining 6 TPU cores for computing the parameter update (these are averaged with an all reduce across the participating cores). Updated parameters are sent to the actor TPU devices via a fast device to device channel as soon as the new parameters are available. This minimal unit can be replicates across multiple hosts, each connected to its own 56 CPU cores and 8 TPU cores, in which case the learner updates are synced and averaged across all learner cores (again via fast device to device communication).

## B.11 JAX-like Pseudocode

In this section, we provide a detailed pseudocode which shows how the learning updates are performed to the manager, option-policy, option-reward and option-termination network parameters. The source code depends on many proprietary software packages and so could not be released. The starting point for this pseudocode in the `update_all_params` fn, which takes in the parameters of the learning agent (theta refers to the option-policy network; eta to option-reward and termination network; manager_params to the manager network) along with their optimiser states. The function also takes a sequence of transitions produced by interacting with the environment. The algorithm, using the sequence of transitions and through this function, applies a number of inner updates to the option-policy parameters. Then, the final inner-loop updated option-policy parameters are used to evaluate the outer-loss, which is used to compute gradients for the option-reward and option-termination network parameters (via a meta-gradient). The final option-policy, option-reward and option-termination parameters are returned along with their updated optimiser states which are used in subsequent training updates. In addition to these updates, the sequence of transitions are used to make a learning update to the manager's policy parameters; the updated parameters and optimiser state are also returned as output.

The `_inner_loss` and `_outer_loss` fns computes the loss function for the option-policy and option-reward (and termination) network respectively. The loss function for option-policy is parameterised by the option-rewards and terminations, while the loss function for the option-policy and terminations is defined with the extrinsic rewards accumulated by the manager's policy. Computing gradients and making learning updates using these loss fns produces the learning updates for those parameters, identical to the Eqns. 1, 5. Similarly, `_manager_loss` and `_manager_update` fns produces learning updates to manager's parameter according to Eqn. 2.

```
def _inner_loss(theta, traj, eta):
    # Inner loss fn: Computes actor-critic style loss using option-policies (theta) &
    # option-rewards and terminations (eta)
    discounts = traj.discount[1:] * discount
    # Unroll the option-policy network to obtain each option-policy and its associated value fn
    # Recall that the option-policy network does not receive the task information
    learner_output = worker.unroll(theta, traj.reduced_observation)
    # Unroll the option-reward and termination network to
    # produce the rewards and terminations for all option-policy
    meta_output = meta.unroll(eta, traj.reduced_observation)
    cumulants = meta_output.cumulants[:-1]
    option_discounts = meta_output.worker_discounts[1:]

    pg_loss, vf_loss, ent_loss = 0., 0., 0.
    # Iterate over the number of option-policies
    for option_policy_idx in range(num_option_policies):
        # Obtain option-rewards (called as cumulants) for the current indexed option-policy
        cumulants_per_worker = cumulants[:, option_policy_idx]
        option_discount_per_worker = option_discounts[:, option_policy_idx]
        actions_per_worker = traj.action_tm1[1:, option_policy_idx]
        cumulants_per_action = jnp.take_along_axis(
                    cumulants_per_worker, actions_per_worker[..., None],
                    axis=-1).squeeze(axis=-1)
        # Masks out the loss for transitions not produced by the current indexed option-policy
        mask_for_update = traj.worker_masks[1:] == worker_idx

        value_output_per_worker = learner_output.value[:-1, option_policy_idx]
        logits_per_worker = learner_output.logits[:-1, option_policy_idx]
        bootstrap_value_per_worker = learner_output.value[1:, option_policy_idx]
        discounts_per_worker = discounts * mask_for_update * option_discount_per_worker
        targets_per_worker = discounted_return_fn(
                        cumulants_per_action, discounts_per_worker,
                        stop_gradient(bootstrap_value_per_worker), 1.)
```

```python
        advantages = targets_per_worker − stop_gradient(value_output_per_worker)

        vf_loss += jnp.square(
                (value_output_per_worker − targets_per_worker) * mask_for_update).mean()
        pg_loss += policy_gradient_loss(
                logits_per_worker, actions_per_worker, advantages, mask_for_update)
        ent_loss += entropy_loss(logits_per_worker, mask_for_update)
    return pg_loss + 0.5 * vf_loss + 0.01 * ent_loss

def inner_loss(theta, eta, traj):
    inner_loss_val = jax.vmap(_inner_loss, (None, 1, None))(theta, traj, eta)
    return inner_loss_val.mean()

def inner_update(theta, theta_opt_state, theta_opt_update, traj, eta):
    # Inner update: Updates option−policies using option−rewards and terminations
    grads = jax.grad(inner_loss)(theta, eta, traj)
    updates, new_theta_opt_state = theta_opt_update(grads, theta_opt_state)
    new_theta = optax.apply_updates(theta, updates)
    return new_theta, new_theta_opt_state

def _outer_loss(theta, traj, manager_params):
    # Outer loss fn: Computes actor−critic style loss using the updated option−policy params (theta)
    # Unroll the option−policy network to obtain each option−policy and its associated value fn
    # Recall that the option−policy network does not receive the task information
    learner_output = option_policy.unroll(theta, traj.reduced_observation)
    # Unroll the manager network to obtain its policy and value fn
    manager_output = manager.unroll(manager_params, traj.observation)

    rewards = traj.reward[1:] − switching_cost * traj.manager_masks[:−1]
    discounts = traj.discount[1:] * discount
    returns = discounted_return_fn(rewards, discounts, manager_output.value[−1]))
    advantages = stop_gradient(returns − manager_output.value[:−1])
    pg_loss, ent_loss = 0., 0.
    # Iterate over the option−policies, computing its outer−loss
    for option_policy_idx in range(num_option_policies):
        # Mask out the loss for transitions not produced by the current indexed option−policy
        mask_for_update = traj.worker_masks[1:] == option_policy_idx
        logits_per_worker = learner_output.logits[:−1, option_policy_idx]
        actions_per_worker = traj.action_tm1[1:, option_policy_idx]
        pg_loss += policy_gradient_loss(
            logits_per_worker, actions_per_worker, advantages, mask_for_update)
        ent_loss += entropy_loss(logits_per_worker, mask_for_update)
    return pg_loss + 0.01 * ent_loss

def outer_loss(eta, theta, theta_opt_state, manager_params, trajs, theta_opt_update):
    # Iterate over the sequence of trajectories,
    # performing a sequence of updates to the option−policy params (theta)
    for j in range(len(trajs) − 1):
        theta, theta_opt_state = inner_update(
            theta, theta_opt_state, theta_opt_update, trajs[j], eta)
    # Use the final option−policy param to evaluate the outer−loss
    outer_loss_val = jax.vmap(_outer_loss, (None, 1, None))(
                            theta, trajs[−1], manager_params)
    # Applies update to option−policy on the last batch of trajectory
    # that was used in evaluating the outer−loss
    theta, theta_opt_state = inner_update(
            theta, theta_opt_state, theta_opt_update, trajs[−1], eta)
    return outer_loss_val.mean(), (theta, theta_opt_state)
```

```python
def outer_update(eta, theta, manager_params, eta_opt_state, theta_opt_state,
                 trajs, eta_opt_update, theta_opt_update):
    # Outer update: M inner updates of option-policies and then use that for computing outer-loss,
    # which is used to obtain meta-gradients for option-rewards and terminations (eta)
    eta_grads, (new_theta, new_theta_opt_state) = jax.grad(outer_loss, has_aux=True)(
        eta, theta, theta_opt_state, manager_params, trajs, theta_opt_update)
    updates, new_eta_opt_state = eta_opt_update(eta_grads, eta_opt_state)
    new_eta = optax.apply_updates(eta, updates)
    return new_eta, new_eta_opt_state, new_theta, new_theta_opt_state

def _manager_loss(manager_params, traj):
    # Manager loss fn: Computes actor-critic style loss for manager's policy
    # Adds a switching cost whenever the manager makes a decision
    rewards = traj.reward[1:] - switching_cost * traj.manager_masks[:-1]
    discounts = traj.discount[1:] * discount
    # Unroll the manager network to obtain its policy and value estimates
    manager_output = manager.unroll(mu, traj.observation)

    returns = discounted_return_fn(rewards, discounts, manager_output.value[-1])
    advantages = stop_gradient(returns - manager_output.value[:-1])
    # Computes policy-gradient, value fn and entropy regulariser losses
    # Apply the policy-gradient and entropy regulariser
    # on the transitions where the manager made a decision
    pg_loss = policy_gradient_loss(manager_output.logits[:-1], traj.worker_masks[1:],
                                   advantages, traj.manager_masks[:-1])
    vf_loss = jnp.square(manager_output.value[:-1] - returns).mean()
    ent_loss = entropy_loss(manager_output.logits[:-1], traj.manager_masks[:-1])
    return pg_loss + 0.5 * vf_loss + 0.01 * ent_loss

def manager_loss(manager_params, trajs):
    # Evaluates the manager's policy performance on the batch of trajectories
    losses = jax.vmap(_manager_loss, (None, 1))(manager_params, trajs)
    return losses.mean()

def manager_update(manager_params, manager_opt_state, manager_opt_update, trajs):
    # Manager update: Updates the manager's policy after
    # evaluating its performance on batch of trajectories
    grads = jax.grad(manager_loss)(manager_params, trajs)
    updates, new_manager_opt_state = manager_opt_update(grads, manager_opt_state)
    new_manager_params = optax.apply_updates(manager_params, updates)
    return new_manager_params, new_manager_opt_state

def update_all_params(theta, theta_opt_state, theta_opt_update, eta, eta_opt_state, eta_opt_update,
                      manager_params, manager_opt_state, manager_opt_update, trajs):
    # Updates option-policies (theta), option-reward and termination (eta) params
    new_eta, new_eta_opt_state, new_theta, new_theta_opt_state = outer_update(
        eta, theta, manager_params, eta_opt_state,
        theta_opt_state, trajs, eta_opt_update, theta_opt_update)
    # Updates manager params
    new_manager_params, new_manager_opt_state = manager_update(
        manager_params, manager_opt_state, manager_opt_update, trajs)
    return new_theta, new_theta_opt_state, new_eta, new_eta_opt_state,
           new_manager_params, new_manager_opt_state
```


| General Hyperparameters | Value |
| --- | :---: |
| Number of environment steps | 200M |
| $n$-step return | 20 |
| Batch size | 32 |
| Number of learners | 1 |
| Number of parallel actors | 200 |
| Learning rate schedule | Constant |

| Manager, Option-Policies | Value |
| --- | :---: |
| Value loss coefficient | 0.5 |
| Entropy coefficient | 0.01 |
| Learning rate | 0.0006 (Atari), 0.0001 (DeepMind Lab) |
| Switching cost | 0.1 (Atari), 0.03 (DeepMind Lab) |
| Number of Options | 5 |
| RMSProp momentum | 0.0 |
| RMSProp decay | 0.99 |
| RMSProp $\epsilon$ | 0.01 |
| Global gradient norm clip | 40 |

| Option-Rewards, Option-Terminations | Value |
| --- | :---: |
| Meta-gradient norm clip | 1 |
| Learning rate | 0.0001 |
| RMSProp momentum | 0.0 |
| RMSProp decay | 0.99 |
| RMSProp $\epsilon$ | 0.01 |
| Inner update steps | 5 |

Table 1: Detailed hyperparameters used by MODAC.

(a) Do the main claims made in the abstract and introduction accurately reflect the paper's contributions and scope? [Yes]

(b) Did you describe the limitations of your work? [Yes]

(c) Did you discuss any potential negative societal impacts of your work? [Yes]

(d) Have you read the ethics review guidelines and ensured that your paper conforms to them? [Yes]

2. If you are including theoretical results...

(a) Did you state the full set of assumptions of all theoretical results? [N/A]

(b) Did you include complete proofs of all theoretical results? [N/A]

3. If you ran experiments...

(a) Did you include the code, data, and instructions needed to reproduce the main experimental results (either in the supplemental material or as a URL)? [Yes]

(b) Did you specify all the training details (e.g., data splits, hyperparameters, how they were chosen)? [Yes]

(c) Did you report error bars (e.g., with respect to the random seed after running experiments multiple times)? [Yes]

(d) Did you include the total amount of compute and the type of resources used (e.g., type of GPUs, internal cluster, or cloud provider)? [Yes]

4. If you are using existing assets (e.g., code, data, models) or curating/releasing new assets...

(a) If your work uses existing assets, did you cite the creators? [N/A]

(b) Did you mention the license of the assets? [N/A]

(c) Did you include any new assets either in the supplemental material or as a URL? [N/A]

(d) Did you discuss whether and how consent was obtained from people whose data you're using/curating? [N/A]

(e) Did you discuss whether the data you are using/curating contains personally identifiable information or offensive content? [N/A]

5. If you used crowdsourcing or conducted research with human subjects...

(a) Did you include the full text of instructions given to participants and screenshots, if applicable? [N/A]

(b) Did you describe any potential participant risks, with links to Institutional Review Board (IRB) approvals, if applicable? [N/A]

(c) Did you include the estimated hourly wage paid to participants and the total amount spent on participant compensation? [N/A]