# OpenReview forum: "Discovery of Options via Meta-Learned Subgoals"
_NeurIPS.cc/2021/Conference — NeurIPS 2021 Poster_

### Official Review · Reviewer_zvKE · 2021-07-10

**Rating:** 6
**Confidence:** 5

**Summary:**

The authors tackle the problem of learning options that would be useful across a range of similar tasks (for example similar mazes with different difficulties). The authors use a meta-gradient approach with appropriately defined parameters and meta-parameters. The paper contains quantitative and qualitative experiments on variants of the FourRooms task and on the more complicated DeepMind Lab (as well as experiments on Atari in the appendix)

**Limitations And Societal Impact:**

Learning options can lead to better interpretability. This could be highlighted.

**Main Review:**

Meta-gradients have been used previously for different settings in RL but this paper seems to be the first to apply it in a HRL setting. The combination of the two could lead to more flexible options that generalize better. The paper is also generally well written and cites many important related work.

That being said, I wish some of the theoretical and experimental details were better motivated.

In particular for the experimental aspect:
1. Although the choice of adding primitive actions to the options set is present in the original paper on options (Sutton et al.), it is not a current practice these days. Importantly, it is not clear whether OC and MLSH are using these primitive actions. It is good to know that options are used the majority of the time, but an ablation analysis looking at the role of adding primitive actions is necessary for me. The reason for this is that it has nothing to do with the meta-gradient approach and it would be enlightening for the community to understand the choice behind adding primitive actions.
2. The way the deliberation cost is not really principled and not motivated theoretically.  The authors don't explain why the switching cost is added to the manager but not to the options themselves. Notice that in Harb et al. 2018 the derivation indicates that it should be used for all updates. Moreover, shouldn't meta gradient to learn temporally extended options, if these options were generally useful? It is not clear why it is also necessary to add a deliberation cost.
3. The first claim of the paper is that "our approach can discover meaningful and diverse temporally-extended options". Personally, when I look at the qualitative results on DeepMind Lab, I am really not sure how the behaviour of each option is useful or interesting. Moreover, could we see similar options emerging from the baselines?
4. The MODAC algorithm, which uses learned rewards, is compared to HRL approaches that don't, that is Option-Critic (OC) and MLSH. I don't find this aspect particularly surprising. I think stronger baselines would be necessary here, at least for the smaller experiments.

In particular for the theoretical aspect:
1. Regarding Equation 3, the authors mention that when the option terminates, there is no bootstrapping. To me this seems simply wrong. Notice that in OC there is always bootstrapping and that the boostrapping is consistent with whether or not there was termination.
2. The derivation of the meta-gradient approach, which is the essential contribution of the paper, is very succinct. Even in the appendix, there is barely any details as to the approximations being considered. As a comparison, please consider the original Meta-Gradient paper (Xu et al.) where many more details are provided (basically the whole of section 1).
3. This leads to the question as to how the choice of the parameters vs hyper-parameters was made. Indeed, according to the original paper on options, the target for updating an option depends on the reward function, the termination function, the policy over options and even the other options' policies. What motivates which functions are considered parameters or meta-parameters?


===== Post-Rebuttal ======

I appreciate the authors' efforts to address the points raised by the reviewers. In particular, I appreciate that the author providing a more competitive baseline where reward shaping is used for learning options. I would suggest to include all the details (such as hyperparameter search) for these additional experiments. Given the authors' efforts in addressing the issues, I will be raising my score to a weak accept even though it would be better to have an updated version of the paper to be convinced.

I think that the authors should be more clear in the approximations introduced in the work. This could be addressed by presenting complete and detailed derivations for their method. Many of the issues raised by the reviewers would then be apparent. As it is, the paper presents the update rules but the derivation of these updates are incomplete, and the answers provided during the rebuttal lead to more questions that remain up in the air.

In particular, both the way bootstrapping is done is not justified theoretically, even though intuitively there could be a motivation. The same goes for the deliberation cost. Why are the authors introducing actions, to then introduce a heuristic to penalize for choosing them? As we are learning options and their length can be changed, it is not necessary to include actions in the first place.

The meta-gradient method already introduces a set of approximations, and at some point it becomes hard to justify why more heuristics are necessary, especially as the empirical performance is not that much better than using a simple flat agent.

**Time Spent Reviewing:**

4

---

> ### Author Response · Authors · 2021-08-10
> **Addressing Reviewer's comments - Reviewer zvKE**
>
> We thank the reviewer for their detailed feedback. We will now address specific comments from the review here.
>
> _**Questions from experiments:**_
>
> 1: HRL baselines (Option-Critic and MLSH) also learn to select among the union of options and primitive actions, identical to our agent. This is described in the 'Baselines' paragraph in Section 3, and also in Section A.2 'Neural network architecture' of the appendix.
> Allowing a HRL agent to select across options and primitive actions is not a contribution of our work. Here we point out two published works that argue for using primitive actions along with options to improve learning performance. Mann, Mannor \& Precup (2015) theoretically study the role of adding primitive actions and proved that using options along with primitive actions allows a planning agent to achieve faster convergence. Jong et al. (2008) empirically studies this design choice of including primitive actions to a HRL agent and clearly show that the agent that can select between options and primitive actions can obtain optimal performance while the agent that only selects options fails. Furthermore, the works by Li et. al. (2019) and Baumli et. al. (2020) also explores HRL approaches that learn by selecting among options and primitive actions. We will include references to all these works to further support our design choice.
>
> 2: The use of switching cost (a negative reward) is based on the intuition of incentivizing the manager to make fewer decisions while trying to maximize rewards from a given task. It does this by discouraging the manager from selecting primitive actions too often as it incurs a lot of switching costs, thereby decreasing its return. The switching cost is used with every update to the manager during the training phase to encourage the agent to select options while they are being discovered.
> 3: The options discovered in the gridworld domain (Fig. 3) specifically support our claim that the options are diverse. Each discovered option leads the agent directly into the rooms where the training tasks were concentrated. In addition to this, each of the options discovered on DMLab domains (Main text Fig. 5; Supplementary Material Fig. 3, 4) lead the agent to different parts of the environment (i.e., produce different behaviors) even when the start position and orientation of the agent was identical for each visualization. The visualizations for DMLab were from the explore_goals_locations tasks, where the agent needs to first explore to find the goal location and then reach that location to receive a reward. In those tasks, it is useful to have options that lead the agent to different parts of the environment as it allows the agent to quickly identify the rewarding location.
>
> 4: Our work focuses on discovering a discrete number of options that can be reused to learn across multiple tasks, particularly when the number of tasks is more than the number of options. For a fair empirical comparison, we considered relevant approaches that also discovered a discrete number of options directly from the extrinsic reward function in a multi-task setting. Also Option-Critic and MLSH seem to be the popularly used methods in hierarchical RL in multi-task domains (e.g: Igl et al. 2020; Frans et al., 2017;).
>
> _**Questions from algorithm:**_
>
> 1:  Equation 3 defines the discounted sum of option-reward achieved until the option terminates, what we call the option-return.  The option return is used to update the option policy (see Equation 1). The semantics for the option-policy is that it is the optimal policy given the option reward and the option termination probabilities (that in our method are discovered using metagradients). This is the usual semantics of an option (as defined in the original Options paper (Sutton et al., 1999)). The fact that the Options-Critic bootstraps based on the value of the termination state is the unusual heuristic they used in their approach.
>
> The motivation behind our approach was to discover options where each of them individually achieve a subgoal. To this end it makes sense to think about options that learn to solve their subgoals without bootstrapping beyond their terminations and the meta-gradient procedure is to come up with option-rewards and terminations (which define the subgoals) to accomplish this.
>
> 2: We provide a detailed derivation in this rebuttal and also point out the approximations made during the derivation.
> Derivation of the Meta-Gradient Update Rule:
> The gradients for option-rewards and terminations are obtained by following a policy-gradient update to maximize extrinsic returns G^M. The policy for the learning agent can be viewed to be a factored policy which is composed of the manager’s policy which selects an option and this selected policy then selects an action which is executed in the environment. The meta-gradient is then computed by differentiating through this factored policy where the parameters of the manager and option-policies are obtained after (atleast) an inner-loop update (In our work, we performed 5 inner-loop updates). The gradient term for the option-rewards and terminations (i.e meta-parameters) are approximate because it ignores the effect of a change in option-rewards and terminations on the manager’s policy. The update term only captures the direct dependence of the meta-parameters on the option-policies.
>
> \begin{align}
>     {\eta^{r^{o_i}}}^{\prime} &= \eta^{r^{o_i}} + \alpha_{\eta} \nabla_{\eta^{r^{o_i}}} \mathbb{E}_{\{ \theta^{o_i} \}, \theta^M, \mathcal{G}} \big[ G^M \big] \nonumber\\
>     &= \eta^{r^{o_i}} + \alpha_{\eta} \mathbb{E}_{\{ \theta^{o_i} \}, \theta^M, \mathcal{G}} \big[ G^M \nabla_{\eta^{r^{o_i}}} \log \big[ \pi^M (o_i | s, g) \pi^{o_i}(a | s) \big] \big] \nonumber \\
>     &\approx \eta^{r^{o_i}} + \alpha_{\eta} \mathbb{E}_{\{ \theta^{o_i} \}, \theta^M, \mathcal{G}} \big[ G^M \nabla_{\eta^{r^{o_i}}} \log\pi^{o_i}(a | s) \big] \nonumber \\
>     {\eta^{r^{o_i}}}^{\prime}&\approx \eta^{r^{o_i}} + \alpha_{\eta} \mathbb{E}_{\{ \theta^{o_i} \}, \theta^M, \mathcal{G}} \big[ G^M \nabla_{{\theta^{o_i}}^{\prime}} \log\pi^{o_i}(a | s)\nabla_{\eta^{r^{o_i}}} {\theta^{o_i}}^{\prime}\big] \nonumber
> \end{align}
>
> \begin{align}
>     {\eta^{\beta^{o_i}}}^{\prime} &= \eta^{\beta^{o_i}} + \alpha_{\eta} \nabla_{\eta^{\beta^{o_i}}} \mathbb{E}_{\{ \theta^{o_i} \}, \theta^M, \mathcal{G}} \big[ G^M \big] \nonumber\\
>     &= \eta^{\beta^{o_i}} + \alpha_{\eta} \mathbb{E}_{\{ \theta^{o_i} \}, \theta^M, \mathcal{G}} \big[ G^M \nabla_{\eta^{\beta^{o_i}}} \log \big[ \pi^M (o_i | s, g) \pi^{o_i}(a | s) \big] \big] \nonumber \\
>     &\approx \eta^{\beta^{o_i}} + \alpha_{\eta} \mathbb{E}_{\{ \theta^{o_i} \}, \theta^M, \mathcal{G}} \big[ G^M \nabla_{\eta^{\beta^{o_i}}} \log\pi^{o_i}(a | s) \big] \nonumber \\
>     {\eta^{\beta^{o_i}}}^{\prime}&\approx \eta^{\beta^{o_i}} + \alpha_{\eta} \mathbb{E}_{\{ \theta^{o_i} \}, \theta^M, \mathcal{G}} \big[ G^M \nabla_{{\theta^{o_i}}^{\prime}} \log\pi^{o_i}(a | s)\nabla_{\eta^{\beta^{o_i}}} {\theta^{o_i}}^{\prime}\big] \nonumber
> \end{align}
>
> 3: In this work, we wanted to explore the possibility of learning option-rewards and terminations through meta-gradients by measuring their utility across multiple training tasks.
> The option-rewards and terminations are neural networks with parameters and there doesn’t seem to be a direct way of learning/estimating them. This is because they do not directly affect the behavior of the agent; they do so only indirectly through the option policies. Thus, the parameters of the option-reward and option-termination networks became the meta-parameters that are discovered by our approach.
> In contrast, there is a direct path for learning the manager’s policy by maximizing extrinsic rewards. Similarly, there is a direct path to learn the option policies for a given option-reward and termination parameters. Thus, the parameters of the manager and option-policies were considered to be parameters in our approach.

---

> > ### Comment · Reviewer_zvKE · 2021-08-27
> > **Followup**
> >
> > Dear authors, thank you for your rebuttal.
> >
> > For the experiments:
> >
> > Regarding Q1 and Q2, my main concern is that the authors additionally introduce different components to their method, which makes it hard to evaluate how much the meta-gradient method helps and how much it relies on these additional choices. Although adding primitive actions can be justified by looking at past literature, adding a deliberation cost in the way that the paper does is not justified theoretically, nor done previously. This deliberation cost then helps to not choose primitive actions too often. Its as if the paper introduces a problem and a heuristic to solve it. Moreover, adding a deliberation cost when the options are using option-reward models seems to be counter-intuitive: if an option needs to be temporally extended, the option-reward model (and option termination) should be able to learn that. As the meta-gradient is the main contribution, I am confused why this simple setting is not investigated.
> >
> > For Q3, I agree that it is possible to infer some significance to the options learned, but what about the baselines? Do they learn something useful too? Even if baselines also learn something useful, presenting these results does not necessary diminish the contribution of the paper, but puts it in better context.
> >
> > For Q4, I noticed in another thread that the authors agreed to add DIAYN to the gridworld experiments, which is much appreciated. If this could be done within the rebuttal period (i.e. by sharing results) I think it could add merit to the proposed method.
> >
> > For the algorithm:
> >
> > For Q1, In Equation 20 of the original options framework (Sutton et al., 1999, Between MDPs and SMDPs), when the option terminates we should be bootstrapping from some value, but here this is not the case in Equation 3 from the submitted paper. This is the same equation used by the OC algorithm. As such, I am confused why the authors call this an unusual heuristic.
> >
> > Thank you for your answers on Q2 and Q3. It would be great to have the full derivation in the paper with more details.

---

> > > ### Author Response · Authors · 2021-08-31
> > > **Addressing Followup Questions**
> > >
> > > > For Q3, I agree that it is possible to infer some significance to the options learned, but what about the baselines? Do they learn something useful too? Even if baselines also learn something useful, presenting these results does not necessary diminish the contribution of the paper, but puts it in better context.
> > >
> > > > For Q4, I noticed in another thread that the authors agreed to add DIAYN to the gridworld experiments, which is much appreciated. If this could be done within the rebuttal period (i.e. by sharing results) I think it could add merit to the proposed method.
> > >
> > > We have added additional results describing the options from DIAYN, OC and MLSH as response to Reviewer JkKg. Hope that those addresses the comments raised here.
> > >
> > >  > For Q1, In Equation 20 of the original options framework (Sutton et al., 1999, Between MDPs and SMDPs), when the option terminates we should be bootstrapping from some value, but here this is not the case in Equation 3 from the submitted paper. This is the same equation used by the OC algorithm. As such, I am confused why the authors call this an unusual heuristic.
> > >
> > > We believe there is a slight confusion here: we want to point out that when learning the policy and value estimates of the manager, the return that is used for this (G^M_t) does bootstrap across option-terminations using the manager’s value estimate (see Equation 4). This is similar to the equation 20 of the original options framework paper.
> > >
> > > For training the option-policies, the motivation behind the approach was to discover option-policies where each of them individually achieve a subgoal (This motivation is based on pre-defined options that were used in Section 7 of the original options paper). Thus, the return (G^o_t; see Equation 3) that is used to train the option-policies (which are computed using the discovered option-rewards and terminations; not the extrinsic rewards) did not bootstrap across their corresponding terminations. We believe that it is unlikely to discover option-policies where each of them achieve a subgoal if the return used for training option-policies bootstrapped across their corresponding terminations. Thus, we do not think this to be an unusual heuristic introduced in our work.
> > >
> > > We will clarify this better in the revised main text.
> > >
> > > > Thank you for your answers on Q2 and Q3. It would be great to have the full derivation in the paper with more details.
> > >
> > > Thank you. We will add the derivation in the main text pointing out the approximations involved there.

---

> > > ### Author Response · Authors · 2021-08-31
> > > **Addressing Followup Questions**
> > >
> > > >  Although adding primitive actions can be justified by looking at past literature, adding a deliberation cost in the way that the paper does is not justified theoretically, nor done previously.
> > >
> > > We would like to point out that a similar deliberation cost (i.e., adding it to the manager's reward) is used in the work of Baumli et al. (2021). The role of the deliberation cost in their work was to add incentive to the hierarchical agent to pick skills over primitive actions.
> > >
> > > We would like to point out that the deliberation cost does not contradict the contribution which is a meta-gradient approach to discovering options. During the training phase, it is merely used to enable the manager to pick temporally-extended options over the primitive actions while those options are being discovered (In early training phase, the options will be poor and the agent might resort to picking primitive actions to solve the task; we view the deliberation cost as a way to avoid this. The approach can discover options only if they get picked during the training phase.).
> > >
> > > Moreover, the deliberation cost is not used in any of the transfer learning experiments. During the transfer learning experiments, the manager's reward is unaffected by the deliberation cost since it is not used. Thus, the learning performances reported in all our experiments (i.e., from transfer learning experiments) are only due to the discovered options, and not due to the deliberation cost.
> > >
> > > Baumli, Kate, David Warde-Farley, Steven Hansen, and Volodymyr Mnih. "Relative Variational Intrinsic Control." In Proceedings of the AAAI Conference on Artificial Intelligence, vol. 35, no. 8, pp. 6732-6740. 2021.

---

> > > > ### Comment · Reviewer_UfaR · 2021-09-01
> > > > **Comment on Followup**
> > > >
> > > > > We would like to point out that a similar deliberation cost (i.e., adding it to the manager's reward) is used in the work of Baumli et al. (2021). The role of the deliberation cost in their work was to add incentive to the hierarchical agent to pick skills over primitive actions.
> > > >
> > > > As far as I understand from the quick read through of the paper, they used this cost during test time and not for the learning of options like you did in this paper. I understand that you did not use such cost during test time and in my opinion this was a correct choice since I do not see why such extra incentive for using options would be necessary for transfer.
> > > >
> > > > > We would like to point out that the deliberation cost does not contradict the contribution which is a meta-gradient approach to discovering options.
> > > >
> > > > However, since you used the cost for the training of options, I would argue that it is somehow related to the main contribution (algorithm/method that learns better options that are better for transfer) because it is an important part of the algorithm. This is because it directly affects options that are learned and from Figure 2 d) it seems that the value of this parameter affects the final performance quite a lot. Furthermore, the comparison is made with methods that do not use such deliberation cost for training (MLSH probably can't use it but OC can).
> > > >
> > > > > During the training phase, it is merely used to enable the manager to pick temporally-extended options over the primitive actions while those options are being discovered (In early training phase, the options will be poor and the agent might resort to picking primitive actions to solve the task; we view the deliberation cost as a way to avoid this. The approach can discover options only if they get picked during the training phase.).
> > > >
> > > > I do not think that poor starting options would necessarily lead to the agent picking primitive actions and not discovering options in this multi-task setting. The final performance with 0 cost in Fig. 2 d) is also not that bad after all. However, I am not certain about this. If this is indeed a problem, I think that you should explicitly mention this as a limitation of the approach.

---

> > > ### Author Response · Authors · 2021-09-02
> > > **Following up**
> > >
> > > Thank you for your followup questions.
> > >
> > > We would like to kindly remind you to read our recent responses to those, along with our rebuttal.
> > > If our rebuttal successfully addresses your concerns we hope that you would take that into account. Thanks!

---

### Official Review · Reviewer_WMS3 · 2021-07-12

**Rating:** 8
**Confidence:** 4

**Summary:**

The author’s propose an options-based hierarchical RL algorithm that uses meta-learned option-rewards and termination to train individual options.
The method is instantiated with an actor-critic RL method and experiments are conducted to test the transfer capabilities of the meta-learned options.

**Limitations And Societal Impact:**

Yes

**Main Review:**

Strengths:
+ Very clear exposition of the method and good writing overall.
+ Detailed appendix with hyperparameter settings and method implementation details.
+ Strong experimental results for options with challenging DeepMind Lab environment and Atari.
+ Visualizations of learned options.

Weaknesses:
+ It would be good to also test against a subgoal based hierarchical RL method (f.e. HIRO)
+ shouldn’t it be $+\kappa^o v^o$ in equation 1 since value should be pushed up if it's underestimating n-step return?
+ Some more motivation/analysis for the benefit of using meta-learned rewards and terminations rather than sub-policies would be helpful. Right now the intuition seems to be that it is less task-dependent. In principle meta-learning could cause option-rewards to become task-dependent just like directly learned options?

Summary:
Novel approach to option discovery using meta-gradients in new way. Strong experimental results and provides a promising direction for options framework and multi-task transfer in RL.

**Time Spent Reviewing:**

3

---

> ### Author Response · Authors · 2021-08-10
> **Addressing Reviewer's comments - Reviewer WMS3**
>
> We thank the reviewer for their detailed feedback and will now address specific comments from the review here.
>
> > Compare against HIRO
>
> HIRO is a method that uses pre-trained options where the semantics for those options are defined by the agent designer (Specifically, the options were defined to reach different locations on the environment). In our work, we are exploring an approach that does not require the agent designer to define semantics for the options, and instead learn them directly from the agent’s experience.
>
> > shouldn’t it be $+κ^o v^o$ in equation 1
>
> The equation in its current form is correct. The reason it seems confusing is because it combines the policy-gradient loss function with the value function minimization loss function, which is not the conventional way of writing down an update equation for an actor-critic architecture. We will modify this update equation to have two separate gradient terms, one for updating the policy with the policy-gradient and another for updating the value estimate with the gradient from the value estimation error.

---

### Official Review · Reviewer_UfaR · 2021-07-15

**Rating:** 4
**Confidence:** 4

**Summary:**

Authors propose a meta-learning algorithm for learning temporarily extended actions within the options framework. In addition to the commonly used option modules defined in the original options framework (high-level policy, terminations, sub-policies) the algorithm also uses an aditional learned option-rewards module ($\{r^{o^i}\}$) to reward sub-policies when they perform actions. The algorithm is designed to extract useful temporarily extended actions from multiple environments by using meta-gradients (backpropagating through gradient). Meta-gradients are taken wrt. parameters of termination functions and option-rewards whereas sub-policies and high-level policy are updated in the inner update. The intuition behind using option-rewards and terminations as meta-parameters is that they should define task-agnostic sub-goals.

Authors evaluate the performance of this new algorithm on one simple and one complex navigation tasks. The main performance metric considered is performance on a new unseen task (after pre-training on many tasks), although authors do show additional visualizations of learned options. Comparison is made with two additional algorithms for learning options  MLSH (designed for multi-task learning) and Option Critic (originally designed for single-task) and a single-task Actor-Critic baseline. In experiments, authors show that proposed algorithm outperforms all three baselines when options are transferred to previously unseen tasks. This suggests that proposed algorithm discovers options with better temporal structure.

**Limitations And Societal Impact:**

Authors do discuss societal impact but do not discuss the limitations or drawbacks of the approach when compared to related methods.

**Main Review:**

The paper introduces an interesting novel method for meta-learning of options. It is clearly written and experiments show that the approach works well on both simple and challenging baseline. However, the method is mostly justified intuitively and I find the theoretical justification of the algorithm and the justification of some design choices that was provided quite weak. It is thus hard to see whether the method is theoretically sound and why it works so well when compared to prior methods. Authors also do not discuss the limitations of the approach. I would thus not recommend accepting the paper unless authors provide a clear derivation of the updates used in the algorithm from the main objective, better justify design choices and discuss limitations.

Originality: The method builds on known techniques but includes a novel contribution (using gradient based meta-learning to learn hierarchical policies). Authors explain the differences and relation to related work although I am not completely sure if there isn't a connection missing between adding a negative reward for options switching and deliberation cost [2].

Quality: Claims are mostly supported by experiments, however, I did not find theoretical analysis of the approach sufficient. From the derivations provided in the paper and the appendix I cannot tell if derived updates are sound and as far as I can tell they seem to be at least a few missing dependencies (see detailed comments). Furthermore, in my opinion, the addition of learned option-reward module needs better justification than what is provided because it is usually not part of the options framework and I do not immediately see why "task reward may be insufficient to discover reusable, task-independent options", why such addition would be necessary and how it would combine with the options framework. I also did not find the limitations of the approach in the paper.

Clarity: The writing is mostly very clear and necessary concepts are well explained. The submission is also well organized.

Significance: The submission tackles an interesting difficult task (learning hierarchical structure in multi-task learning) and according to experiments outperforms well-known baselines (also on non-trivial task). Additionally, it presents a novel idea: using gradient based meta-learning in combination with the options framework to learn such structure. I believe that this idea and its realization would be a contribution to the research community.

**Detailed Comments\Questions:**
- The option-reward module is only shortly introduced in line 95 with the intuition (line 105) that they define an option's subgoal and implicitly defines the option-policy (line 79). However, the connection of this module with the options framework is a bit unclear. I think that authors should provide an explanation why these options-rewards define sub-goals and show what kind of option-reward functions (subgoals) are learned in four rooms environment. I think this idea might be somehow connected to the Option Keyboard [3].

- I think that addition of "switching cost $c$" could be connected to the Deliberation cost [2]. Doesn't this interfere with the objective/update equations?

- From Eq. 3 and the learning algorithm, I understand that $r^o$ are not connected to real rewards in any way. I do not see the motivation for $G_t^o$ or why it is sensible to interpret terminations as a discount. Why does it make sense to interpret $\beta, r^o$ and $v^o$  as discounted returns? Why isn't normal discounting used? Since $\beta, r^o$ and $v^o$ are all meta-learned isn't $G_t^o-v^o(s_t)$ just an arbitrary

- The objective and following derivation in the appendix (line 75) does not seem correct since it does not take into account the dependency of sampling distribution $p(G_t^M|\{\theta^o_i\},\theta^M, \{\eta^{\beta^o_i}\})$ on terminations $\beta$. Following line 81 in the appendix, why is only a single $\log\pi^o$ inside both expectations? As far as I can tell , it should be whole $p(G_t^M|\{\theta^o_i\},\theta^M, \{\eta^{\beta^o_i}\})$ decomposed into multiple components. Furthermore, I think that is not sufficient to differentiate through ${\theta^o}'$ since ${\theta^M}'$ (after inner updates) also depends on termination parameters (through sampling distribution as was explained in [1]) . Updates in Eqs. 1,2,5 are also written in terms of single samples but they are actually estimates of expectations which makes a difference in meta-learning setting.

- In line 55 authors claim that all options directly optimizing the same (main task) reward may not be sufficient to discover reusable task-independent options. Why/In which settings is this a case?

- In Fig 2 c) you show average return. However in such environments where reward for reaching a goal is one, one usually optimizes for discounted return (since return is always 1 if agents finds the goal). Is there any specific reason why you are showing return instead?  Can you provide plots for discounted return?

- Could you add the visualization of terminations and learned option-rewards module for four rooms task?

Typos and minor details:
- line 32 cite
- line 63 use citations with MLSH and OC? (they are used later instead)
- use $\theta'$ on the left side in Eqs 1 and 2
- introduce L in algorithm box
- line 166 Coagent Networks citation
- Fig 2b text is very small
- line 269 sight instead of site
- line 299 as far as I remember Mankowitz et al. 2016 does not require hand-designed temporal abstractions

[1] DiCE: The Infinitely Differentiable Monte Carlo Estimator (Foerster et al. 2018)
[2] When Waiting is not an Option : Learning Options with a Deliberation Cost (Harb et al. 2017)
[3] The Option Keyboard: Combining Skills in Reinforcement Learning (Barreto et al. 2019)

===== Post-Rebuttal ======
After the discussion with authors and other reviewers I decided to keep my score unchanged. Even though I do appreciate the effort that authors put into providing an additional baseline, I do not think that the issues related to theoretical justification of the method were properly addressed. While I realize that the work is mostly empirical, I do think that the empirical adjustments to the method should be better motivated and their effects on theoretical algorithm should be discussed in more detail especially since the performance of the algorithm is not that much better than the one of the single-task agent.

**Time Spent Reviewing:**

8

---

> ### Author Response · Authors · 2021-08-10
> **Addressing Reviewer's comments - Reviewer UfaR**
>
> We thank the reviewer for their detailed feedback. We will now address specific comments from the review here.
>
> > option-reward module is only shortly introduced...
>
> The options framework does not define a way to discover option-policies. In our work, we view the option-policies as those that achieve specific subgoals, which are obtained by learning to maximize their own intrinsic reward functions (i.e., option-rewards). The option-rewards are functions of states and actions, making it difficult to visualize them in a way to make conclusions about them. An alternate way to understand what the option-rewards and terminations are being discovered is to look at the resulting option-policies as they are produced by learning to maximize their corresponding rewards and terminations; which is shown for the gridworld and DMLab domains in the main text and supplementary material. However, in our revision, we will add the option-reward and termination visualizations to the supplementary material.
>
> > addition of "switching cost c". Doesn't this interfere with the objective/update equations?
>
> To understand the role of switching costs, first consider the case where there are no options and so all policies have to choose an action at every time step and pay the switching cost and thus in that case switching costs will effectively play no role. When there are options in addition to primitive actions, then depending on the switching cost the manager is incentivized to select options preferentially. This may lead to overall suboptimal behavior if the discovered options are not perfect for the task -- indeed we expect this to be true in general. That said, in our empirical work we treat the switching cost as a hyperparameter and show that with reasonable choices we get a benefit from discovering and using options.
>
> > From Eq. 3 and the learning algorithm…
>
> The $G_t^o$ is the target estimated by the value function $v^o$ for the task implicitly defined by meta-learned reward function $r^o$ and the meta-learned discount (as a function of state) for each option. We use the $G_t^o$ to learn the option policy just as in any regular RL task (specified by a reward function and a usually state-independent discount) and we learn a value function to learn the policy. Discounts and terminations are equivalent ways of defining options (see Sutton et al., 2011). A quick intuition for this is the following. A discount factor of $\gamma$ at state s is equivalent to terminating with probability (1-$\gamma$) and continuing with probability \gamma.
>
> In the outer-loop updates, it is correct in that they are only approximate as they do not capture the dependency of the meta-parameters (option-rewards and terminations) on the sampling distribution induced by the manager’s policy and only captures the direct dependency on the option-policy parameters. This will be clarified in our main text. We would like to point out that this is also the case with existing meta-gradient literature: in the outer-loop, meta-gradients are computed only through the parameters that are defined as explicit functions of meta-parameters while the remaining parameters are kept fixed (e.g: in Xu et al., (2018) and Zheng et al., (2018) the gradients through the sampling distribution are left out and not part of the outer-loop update). Also, thank you for pointing out the possibility of computing the exact gradients through the manager’s parameters based on the DICE framework. We will explore this in our future work.
>
> The update terms shown in the main text are assuming single samples for simple description. In the algorithm (main text) and in the pseudocode (supplementary material), we do point out that the updates are indeed made from samples obtained across multiple tasks. We use the same setup for the hierarchical baseline agents that are compared against in our experiments.
>
> Sutton, R. S., Modayil, J., Delp, M., Degris, T., Pilarski, P. M., White, A., & Precup, D. (2011, May). Horde: A scalable real-time architecture for learning knowledge from unsupervised sensorimotor interaction. In The 10th International Conference on Autonomous Agents and Multiagent Systems-Volume 2 (pp. 761-768).
>
> > Claim that all options directly optimizing the same (main task) reward may not be sufficient…
>
> The usefulness of the options in the approaches referred to in lines 54-55 are measured directly from a single task’s reward and not based on how useful they are across multiple tasks. This is different in the case of MODAC: the option-rewards and terminations are discovered by measuring their utility on a new batch of trajectories which is obtained from a task that is distinct from the one used during the inner-loop updates. Thus, the option-rewards and terminations should yield option-policies that are useful across multiple tasks, which as per our hypothesis should lead to them generalising better to unseen tasks.
>
> > Average return vs discounted return
>
> Consider the line of work on Atari games. There the performance we care about is that of the undiscounted score. Yet, most if not all the algorithms/agents use a discount factor of 0.99. In the empirical results, however, they show undiscounted scores. This is because the use of a discount factor less than 1 is an algorithm parameter, NOT an element of the true performance we care about. Indeed, it has been standard practice in RL to use a discount factor less than one to promote the stability of the RL algorithm because we don’t have stable and very-high-performance undiscounted RL algorithms at this point. Our results are in the same standard practice. We will make this clear in the revision.
>
> > Visualization of option-rewards and termination
>
> We will add visualizations from option-terminations and option-rewards to the supplementary material.

---

> > ### Comment · Reviewer_UfaR · 2021-08-31
> > **Followup**
> >
> > Dear authors, thank you for your rebuttal.
> >
> > > The options framework does not define a way to discover option-policies. In our work, we view the option-policies...
> >
> > I think that I intuitively understand how option specific reward functions can induce a sub-policy. However, I still think that since this is not a common part of the option framework, it would be beneficial for the paper to either include a reference to prior work that uses similar idea or discuss the limitations/consequences/assumptions of using such intrinsic-rewards module.
> >
> > > To understand the role of switching costs, first consider the case where there are no options and so all policies...
> >
> > I am aware that switching cost is also used in other works on options to encourage options that are temporarily extended. However, as was also pointed out by Reviewer zvKE, using this cost only in managerial update likely introduces bias and when this cost is combined with other empirical decision choices it becomes harder to quantify the added value of meta-gradient.
> >
> > > The usefulness of the options in the approaches referred to in lines 54-55 are measured directly from a single task’s reward...
> >
> > While this is true for Option-Critic, MLSH is trained on multiple environments so the options have incentive to be useful on multiple tasks. I agree that by using another task for post-update trajectories MODAC is different from both of these but it is perhaps better to explain the difference with MLSH separately since it uses multi-env setting.
> >
> > > Consider the line of work on Atari games. There the performance we care about is ...
> >
> > I do think that results with discounted and undiscounted return will be quite similar. However, I was referring to the fact that the agent receives 0 reward unless it reaches the goal (1 reward). In such case agent that reaches the goal in 500 steps would probably get the same undiscounted return as agent that reaches the goal in 10. Thus it might be more informative to include plot with discounted return.

---

> > > ### Author Response · Authors · 2021-08-31
> > > **Addressing Followup Questions**
> > >
> > > Thank you for your followup comments. We will address each one in the following paragraphs:
> > >
> > > > I think that I intuitively understand how option specific reward functions can induce a sub-policy. However, I still think that since this is not a common part of the option framework, it would be beneficial for the paper to either include a reference to prior work that uses similar idea or discuss the limitations/consequences/assumptions of using such intrinsic-rewards module.
> > >
> > > Our work is not the first in using option-specific reward functions to induce an option. Similar ideas were explored by van Seijen et al. (2017), Riedmiller et al. (2018) (where the option-specific rewards were pre-defined by the agent designer). We will include those citations to discuss this design choice and also add discussions on using option-specific intrinsic rewards.
> > >
> > > van Seijen, Harm, Mehdi Fatemi, Joshua Romoff, Romain Laroche, Tavian Barnes, and Jeffrey Tsang. "Hybrid reward architecture for reinforcement learning." In Proceedings of the 31st International Conference on Neural Information Processing Systems, pp. 5398-5408. 2017.
> > >
> > > Riedmiller, Martin, Roland Hafner, Thomas Lampe, Michael Neunert, Jonas Degrave, Tom Wiele, Vlad Mnih, Nicolas Heess, and Jost Tobias Springenberg. "Learning by playing solving sparse reward tasks from scratch." In International Conference on Machine Learning, pp. 4344-4353. PMLR, 2018.
> > >
> > > > While this is true for Option-Critic, MLSH is trained on multiple environments so the options have incentive to be useful on multiple tasks. I agree that by using another task for post-update trajectories MODAC is different from both of these but it is perhaps better to explain the difference with MLSH separately since it uses multi-env setting.
> > >
> > > Thanks for pointing this out. We will separately discuss MLSH as it was also introduced to work in a multi-task setup and not combine its discussion with OC.
> > >
> > > > I do think that results with discounted and undiscounted return will be quite similar. However, I was referring to the fact that the agent receives 0 reward unless it reaches the goal (1 reward). In such case agent that reaches the goal in 500 steps would probably get the same undiscounted return as agent that reaches the goal in 10. Thus it might be more informative to include plot with discounted return.
> > >
> > > Thanks for clarifying this. We want to point out that we also measured the number of steps taken by each agent to complete a task during the transfer learning experiment and the qualitative results across the agents remained similar to the avg. reward curves shown in Figure 2c (i.e., MODAC produced faster learning performance in terms of the number of steps taken compared to MLSH, OC and Flat). We will include this plot into our revision to clarify this comment.

---

> > > ### Author Response · Authors · 2021-08-31
> > > **Addressing followup comment on switching cost**
> > >
> > > > I am aware that switching cost is also used in other works on options to encourage options that are temporarily extended. However, as was also pointed out by Reviewer zvKE, using this cost only in managerial update likely introduces bias and when this cost is combined with other empirical decision choices it becomes harder to quantify the added value of meta-gradient.
> > >
> > > We have addressed this in our response to Reviewer zvKE and hope that it addresses the concern raised here.
> > >
> > > As mentioned in that response, we want to point out that the deliberation cost is used only during the training phase and is not used in the transfer learning experiments. Thus, the learning performances reported in all our experiments (i.e., from transfer learning experiments) are only due to the discovered options.

---

### Official Review · Reviewer_tdpW · 2021-07-16

**Rating:** 8
**Confidence:** 4

**Summary:**

The authors introduce MODAC, a method that relies on meta-gradients for option discovery in hierarchical multi-task settings. The authors extend the options framework to the multi-task setting and use meta-gradients learning to discover task-independent options that can be successfully re-used across tasks. Their architecture is based on the standard hierarchical agent from Sutton et al. (1999) where a manager, that is conditioned by the current state and the task to achieve, can choose between a set of primitive actions and temporally extended options. Adding primitive actions to the action space of the manager adds flexibility and expressiveness and matches the original hierarchical framework. MODAC also implements option-policies, option-rewards and option-terminations. The training corresponds to two nested loops. An inner loop updates the manager to maximize the sum of discounted environment rewards and the option policies to maximize the discounted sum of option rewards. An outer loop evaluates the updated manager and option-policies and updates the option-rewards and option-terminations by back-propagating through the inner loop updates. This training setting enables MODAC option policies to optimize for different objectives that are parameterized by task-independent option-reward and termination that are discovered to be directly useful across many tasks. The authors evaluate MODAC in a grid world environment and in  DeepMind lab navigation tasks. They illustrate that MODAC does not only call primitive actions but also makes use of the options and that the discovered options are diverse and useful for the task at hand. They also compare MODAC to two baselines and show that MODAC outperforms them either in terms of asymptotic performance or sample efficiency or both.

**Limitations And Societal Impact:**

This work provides a method to discover options in multi-task HRL. Thus, it does not add limitations or potential negative societal impact to existing hierarchical reinforcement learning methods.

**Main Review:**

I warmly thank the authors for their work.

Soundness of the claims, significance and novelty of the contribution, and relevance to the NeurIPS community:

- Discovering options, in an automated manner and without any priors, that can be reused to solve distributions of tasks, has been a long-standing goal in RL and I believe this method brings a step forward in that direction.  Using meta-gradients to learn option-rewards and terminations that are themselves used to train task-independent option-policies is novel and seems to work well.

- Allowing the manager to call primitive actions in addition to options is a strength of the method and the authors demonstrated the interest in doing so through examples and during the experimental study.

- The experimental study is thorough and well illustrates the strengths of the method as well as validates the claims made in the paper.



Limitations of this work:

- In part 1, I think the authors should briefly introduce the notations relative to MDPs. For instance, the state space S and the goal space G are never properly introduced. It would also be great if the authors may give more information about the nature of the different spaces. For instance, it is never mentioned that if the action space is discrete or continuous. Also it not clear to me what is the nature of the goal space G. Is it assumed to be a sub-part of the state space or a set of possible tasks?

- In part 1, the number of options K is introduced lately and almost implicitly when the authors describe the outer loop of the algorithm. I would recommend introducing it sooner in the part.

- It would be interesting if the authors could discuss to which extent MODAC could be extended to continuous action spaces and assess its performance on traditional HRL benchmarks such as the ones considered in HAC.

**Time Spent Reviewing:**

4.5

---

> ### Author Response · Authors · 2021-08-10
> **Addressing Reviewer's comments - Reviewer tdpW**
>
> We thank the reviewer for their detailed feedback and will now address specific comments from the review here.
>
> > introduce the notations relative to MDPs
>
> Thanks for the feedback. We will add a section on clearly defining the terminologies and notations used in the main text.
>
> > number of options K is introduced late
>
> Thanks for pointing this out. We agree that our approach does discover a discrete number of options and this detail was introduced only in the algorithm section, which is late for the reader. We will modify the text to make this clear in Section 3 where the agent and algorithm is introduced.
>
> > Discuss MODAC extension to continuous actions
>
> Exploring continuous action spaces is an interesting future direction. We will add a discussion on the feasibility of this to continuous action spaces. We think that this extension should be fairly straightforward.

---

> > ### Comment · Reviewer_tdpW · 2021-08-31
> > **Thank you**
> >
> > Thank you for your answers. I acknowledge that I have also read the other reviews and discussions. I think this work is very promising and hope it will make it to the conference. I am keeping my score of 8 unchanged.

---

### Official Review · Reviewer_JkKg · 2021-07-17

**Rating:** 6
**Confidence:** 4

**Summary:**

This paper develops an approach for option discovery using meta-gradient descent to optimize a set of option networks. These option networks are then used to define the reward, and termination functions. The option policies are learned through normal RL in a separate option-network which is used to define options for a network which learns the behaviour policy using options and the base actions. They restrict their setting to that of multi-task reinforcement learning, where they sample from distributions of tasks with shared underlying regularities. They test their approach on several environments.

**Limitations And Societal Impact:**

I think the limitations of the paper are not well discussed. Specifically the sample inefficiency. I discuss this lightly in my review, but feel this should be addressed with some more comparisons.

**Main Review:**

I think this is a very natural instantiation of the meta-learning for discovery idea proposed by [Veeriah et. al. 2019] to the option discovery problem. Many of the algorithms features are shared, and are directly applicable to the option discovery framework. I believe the ideas and paper have a clear place in the literature, but I'm concerned with the empirical evaluation as is. I'm leaning towards acceptance, but would like some of my points below to be seriously considered in the final revision thus I'm setting my score to a weak reject. I will raise my score if the following are addressed: (S2, S4, Q2, Q3, Q4, S6, S7)

## Introduction, Background, and Algorithm design

S1. I think temporal abstractions, temporally-extended actions, and options are all used interchangeably (which they all are basically interchangeable). This leads to some clarity issues. I think you should choose a term and use it.

S2. I like your hypothesis as it is stated. But I think it could be made more clear and more inline with the rest of the paper. I think the problem is in the focus on discovery in the hypothesis. Instead, maybe "If a temporal abstraction is /useful/ across /many training tasks/, then this abstraction captures regularities shared by the training tasks leading to a higher likelihood of being useful and reusable in new, /previously unseen tasks/." What I like about this rephrasing is it makes clear intuitions about why options are important, why you think the multi-task setting is the right place to test new algorithms, and gives a clear indication that what we are studying is the temporal abstraction (option). The discovery of such things is not important to the hypothesis itself.

Also within this hypothesis there are several terms which are ambiguous at stating. While they are later defined, having clear definitions at the stating of the hypothesis could be beneficial. This might mean moving the statement of the hypothesis to later sections after enough framework is built up.

S3. I'm not sure I would consider Coagent Networks as a technique for discovering options. I'm fine to disagree with that point, but the mention at line 166 should be cited appropriately (i.e. likely [Thomas, 2011] and [Kostas, 2019]).

S4. I think a problem statement would add clarity to the presented ideas. This will also remove ambiguity on why some baselines are included over others later in the paper.

## Empirical Evaluation

Before I get into the statements and questions, I would like to focus on the baselines used for the empirical evaluation. Several notable algorithms are missing: [Machado et al. 2017 & 2018], [Harutyunyan et al. 2019], [Gregor et al, 2016], and [Eysenbach et al. 2018]. Now I don't think all these algorithms should be included (especially because the amount of compute is quite large), but I'm not sure I get the reasoning behind not including an unsupervised approach. The claim is because they are motivated by single-task RL they shouldn't transfer to the multi-task setting. I'm not sure I agree with this, and it is possible because they are learning options through an unsupervised mechanism (some without knowledge of the goal states) they could discover options which are useful for many tasks. [Gregor et al, 2016] even mentions as such in section 3. I believe this is the piece of the empirical evaluation that is difficult for me to accept, and is a large flaw in the paper (and a flaw in how option algorithms seem to be evaluated in general).

One point of concern I've noticed with the results presented in the four rooms environment: the amount of samples used for training all the competitors is significant. Given other option discovery approaches such as [Machado et al. 2018] and [Harutyunyan et al. 2019] use considerably less (on the order of 1000 episodes). If we assume worst case in [Harutyunyan et al. 2019] which is 5000 steps per episode (figure 4), that is at most 5mil samples, which is 100x less than what is used to train your networks. In reality [Harutyunyan et al. 2019] use far less than this. Maybe this is a property of the multi-task setting, but it is hard to know without a comparison.

Q1. What do you mean by usefulness for transfer? Do you mean how much they are used by the manager for testing tasks? Or something else? How are you measuring this?

Q2. What is the motivation for the input space for the four rooms environment? It seems like a lot of information about the whole environment is being passed through the network. Something like only the tabular representation (not the full map of the environment) or the (x,y) coordinates of the agent would be a more reasonable choice.

S5. The plots in figure 2 are quite difficult to read. Maybe it would be better if they were a 2x2 grid rather than 4 plots across the top.

Q3. What are the shown confidence intervals?

Q4. How many runs were used to tune the hyperparameters?

S6. I think including details on hyperparameters in the main paper is an important addition. I think we should always strive to be able to replicate the experimental setup entirely from the details presented in the main paper. Specific values of hyperparameters, and details about frameworks can definitely be pushed to the appendix.

S7. On the number of runs used for the empirical results. While your results look significant given the confidence intervals, I'm not very convinced given the insights from Deep Reinforcement Learning that Matters by Henderson. While computationally it may not be feasible to run more runs for this submission cycle, I think the paper would benefit from this. At the very least, providing confidence intervals using some of the methods discussed by Henderson et al. would be beneficial (say bootstrap). I have this issue for both the four rooms domain and the deep mind lab domain (where number of runs isn't reported).

#### Post response

From my conversation with the authors, I think I will increase my score to a 6 for now. There are still concerns over the bias introduced by the deliberation cost as it relates to the meta update, that I will have to think further on.

As of now, I view this conversation as I view most arguments between theory and empiricism. Because this paper relies much more heavily on being empirical rather than theoretical, these issues might not be problematic for this paper as it is.

**Time Spent Reviewing:**

4

---

> ### Author Response · Authors · 2021-08-10
> **Addressing Reviewer's comments - Reviewer JkKg**
>
> We thank the reviewer for their detailed feedback. We will now address specific comments from the review here.
>
> **_S1:_** Thanks for pointing this out. We will use temporally-extended actions the first time we refer to options and then use the word options thereafter.
>
> **_S2:_** Thanks very much for this valuable feedback. We agree that it was a mistake to state the hypothesis with the notion of discovery in it. We will rephrase the hypothesis along the lines you suggest to be: “If a temporal abstraction is useful across many training tasks, then this abstraction captures regularities shared by the training tasks leading to a reasonable likelihood of it being useful and reusable in new previously unseen tasks.” for the very reasons you suggested the change.
>
> **_S3:_** We cited the Coagent policy gradient paper because it claims that its methods for training stochastic hierarchical networks could extend to discover options. However, we agree that it isn’t a great reason to cite this paper and we will remove it.
>
> **_S4:_** We will reorganize the main text as per the previous style comments and will include the following problem statement for the discovery method we contribute.  Our aim is a “method to discover options that are useful in terms of helping an agent achieve high cumulative reward in multiple training tasks.”
>
> >“Baselines used for the empirical evaluation. Several notable algorithms are missing: [Machado et al. 2017 & 2018], [Harutyunyan et al. 2019], [Gregor et al, 2016], and [Eysenbach et al. 2018]”
>
> Our starting point in this paper is our hypothesis that temporal abstractions useful across multiple training tasks are likely to be useful in previously unseen tasks. Thus, our objective is to develop methods capable of finding temporal abstractions that are *useful* on the training tasks, i.e., to discover options based on the rewards achieved in multiple training tasks. Thus, we chose not to compare against unsupervised methods developed for learning options without regard to reward (however, we did compare against an extension of the single-task Option-Critic method to the multi-task setting). Now, one could of course compare against unsupervised methods but it is unclear what we would learn from such comparisons. There are likely to be some domains where unsupervised methods find options useful for the unseen tasks as well as some domains where unsupervised methods fail to find useful options. It is not our claim that unsupervised methods always fail to find useful options. What we are contributing is a novel method for discovering options that are useful in terms of reward across multiple training tasks and we compare directly against alternative state of the art approaches for doing that.
>
> >[Harutyunyan et al. 2019] use far less samples
>
> The gridworld experiments in Machado et al. (2018) and Harutyunyan et al. (2019) used tabular representations and were single tasks. We conjecture that the number of samples for training MODAC and the baselines are significantly higher because of our choice of using a 3D representation as observation from the gridworld and also because of our multi-task setting.
>
> _**Response to Q1:**_ By usefulness of options during transfer, we mean how much the manager uses them during the test tasks. This is measured by keeping counts for each option and primitive action on how often they are selected by the manager during the episode. Note that when the manager selects an option, the manager will not select again until the option terminates.
>
> _**Response to Q2:**_ In essence the top-down vision view of the gridworld contains exactly the same information that a standard gridworld contains. One channel indicates positions of the walls, another the position of the agent, and the third the goal location. We did this instead of using a lookuptable simply to reuse our code that employs a neural network with convolutional layers. We realize this is unnecessary in gridworlds of the size we employed but it was convenient and we don’t expect the results to be different because of this choice.
>
> _**Response to S5:**_ We will modify Figure 2 from the main text to make it more accessible to the reader. If space allows we will follow the reviewers suggestion.
>
> _**Response to Q3:**_ The confidence intervals show the standard error which are computed from the learning curves obtained from 6 independent runs of the experiment.
>
> _**Response to Q4:**_ We used 3 independent runs to tune the hyperparameters for each agent. For both the hierarchical and the flat agents (MODAC, Option-Critic, MLSH and Actor-Critic) we tuned the entropy regularizer coefficient and the learning rate. The hierarchical agents also used a switching cost which was treated as a hyperparameter.
>
> In our gridworld experiments, we used the tuned hyperparameters for each agent to report their learning performances.
>
> In DMLab, we tuned the hyperparameters for each agent on the explore_goal_locations task and used those tuned hyperparameters to report learning curves from the four DMLab tasks.
>
> In Atari, we again tuned the hyperparameter for each agent on MsPacman and used those tuned hyperparameters to report the results from the four Atari games.
>
> Specific ranges of values for the hyperparameters that were considered in a gridsearch are reported in Section B.6 of the supplementary material.
>
> _**Response to S6:**_ Thanks. We will add the details on what the hyperparameters are and the procedure for their tuning in the main text. Their specific values are and will be provided in the supplementary material.
>
> _**Response to S7:**_ The number of runs for each gridworld and DMLab experiment are reported in Line 226 and Line 261. We used 6 independent runs to report the learning curves from our experiments. We will add learning curves from at least 4 more independent runs for each of our results in a revised version along with their confidence intervals (as suggested by Henderson et al.).

---

> > ### Comment · Reviewer_JkKg · 2021-08-19
> > **More comments**
> >
> > Thank you for the response! Most of the answers are sensible and I appreciate the extra info/changes to the paper, especially when it comes to details about the experiments. I must have missed some details, so making sure these are clearly stated is important. I still have concerns about s4 and q2. But I think some more discussion might help. See below.
> >
> > S4:
> > What might we learn from such a comparison?
> > As far as I can tell, the overall problem statement is to discover options which are useful across tasks. While you focus on a sub-part of this problem (i.e. options useful in multiple tasks will be useful in other tasks), the main objective of "find useful options which are task agnostic" remains the same. I agree, the comparison would not be a typical "method x performs better than method y" as we see in modern ML/RL. Instead it would give room to understand how this new objective for finding options would compare to other ways of discovering task agnostic options (i.e. unsupervised). This comparison could likely be more discussion driven with a comparison of the options derived and a look at how the performances compare on the two distribution of tasks for your method (i.e. training/testing). It would likely not require a very rigorous comparison like the later experiments.
> >
> > Even though the assumptions you use to make your algorithm are different to the unsupervised objectives the overall goals are well aligned, and there is a lingering question this paper should justify. "Why should we go this direction and not us unsupervised techniques?" This question is core to the motivation behind this paper, and I think it is important to at least discuss a bit more in the paper beyond what minimal discussion currently exists.
> >
> > Q2:
> > This is quite an unsatisfactory answer. Especially as there is more information provided to the agent (i.e. the wall channel) than in a strictly tabular sense. Couldn't you instead just use the the channel for the agent's position with dense layers to test on something that is a bit more true to form of the original input space of a grid world?

---

> > > ### Author Response · Authors · 2021-08-24
> > > **Addressing additional comments**
> > >
> > > Thank you for elaborating on your previous comments, this will help us to improve our submission in our revision.
> > >
> > > > S4: What might we learn from such a comparison? As far as I can tell, the overall problem statement is to discover options which are useful across tasks. While you focus on a sub-part of this problem (i.e. options useful in multiple tasks will be useful in other tasks), the main objective of "find useful options which are task agnostic" remains the same. I agree, the comparison would not be a typical "method x performs better than method y" as we see in modern ML/RL. Instead it would give room to understand how this new objective for finding options would compare to other ways of discovering task agnostic options (i.e. unsupervised). This comparison could likely be more discussion driven with a comparison of the options derived and a look at how the performances compare on the two distribution of tasks for your method (i.e. training/testing). It would likely not require a very rigorous comparison like the later experiments. Even though the assumptions you use to make your algorithm are different to the unsupervised objectives the overall goals are well aligned, and there is a lingering question this paper should justify. "Why should we go this direction and not us unsupervised techniques?" This question is core to the motivation behind this paper, and I think it is important to at least discuss a bit more in the paper beyond what minimal discussion currently exists.
> > >
> > > Thank you for pointing out the usefulness of making a comparison with unsupervised option-discovery approaches. While this comparison would not be an apples-to-apples comparison with our approach, we do agree that it would address the motivation behind the problem setup of discovering options from multiple tasks and transferring them to unseen tasks drawn from a similar distribution. We will add comparisons with unsupervised option discovery approaches, specifically, DIAYN (Eysenbach et al., 2018) on the gridworld and discuss the results in our revision.
> > >
> > > > Q2: This is quite an unsatisfactory answer. Especially as there is more information provided to the agent (i.e. the wall channel) than in a strictly tabular sense. Couldn't you instead just use the the channel for the agent's position with dense layers to test on something that is a bit more true to form of the original input space of a grid world?
> > >
> > > We understand that the top-down observation from the gridworld does provide additional information to the learning agent such as the walls in the gridworld. While this was done to have a single codebase that works across different experiments, the choice of using the agent’s position could also have been used which might have been simpler.
> > >
> > > We would still like to point out that the comparison is fair wrto the baseline agents since all of them use the same channel-based observation from the gridworld.

---

> > > ### Author Response · Authors · 2021-08-31
> > > **Options discovered by MLSH, OC and MLSH**
> > >
> > > ## Options discovered with DIAYN:
> > >
> > > We implemented DIAYN on the gridworld domain that was presented in Section 2.1.
> > >
> > > We trained DIAYN to learn 4 discrete options, since all the hierarchical methods in this experiment learned 4 options. The discriminator takes as input the initial and final states reached by an option and outputs the probability of the option that is responsible for producing the transition. This discriminator is trained to maximize the mutual information between the options and the initial and final states resulting from their execution. We used DIAYN’s open-sourced code for this experiment. The length of each option was set to be 5 (same as the tuned hyperparameter for MLSH). Also, the number of training steps used by DIAYN is the same as those used for all the hierarchical methods in this experiment, which is 10M.
> > >
> > > We observed that each option learned to lead the agent in each cardinal direction relative to the start state: Option 1 moved the agent to the right from its start state (i.e., picks the move-right action). Options 2, 3, and 4 moved the agent in the down, up and left directions by picking the corresponding primitive action until the option was terminated. The options discovered from DIAYN did not have a concrete destination state (i.e., subgoals) for each option similar to the ones we observed with MODAC (as shown in Figure 3). The transfer learning performance of these options were comparable to the Flat baseline agent (Figure 2c). We will include these visualizations, results and the above discussions in our revision.
> > >
> > > We think that adding this comparison with DIAYN justifies the overall motivation behind our approach which is to discover options from multiple training tasks (when the agent has access to such training tasks) as opposed to using unsupervised discovery approaches.
> > >
> > > ## Options discovered by Option-Critic and MLSH:
> > >
> > > We also looked at the options that were discovered by Option-Critic and MLSH. We observed that all 4 options discovered by these methods were quite similar and the options led the agent to the center of the closest room depending on the agent’s start state. The options discovered by these methods seem to have multiple destination states (i.e, subgoals) depending on the agent’s start state. In comparison, MODAC learned an option to lead the agent into the center of a room from any starting state in the gridworld (Figure 3). This could potentially explain the difference in transfer learning performance between the options discovered by Option-Critic, MLSH and MODAC (Figure 2c). We will add the option visualizations from Option-Critic and MLSH in our revision.

---

> > > > ### Comment · Reviewer_JkKg · 2021-09-02
> > > > **Thank you for the comparison**
> > > >
> > > > Thank you for the comparison! This is really insightful and should be expanded on in any later revisions of this paper! I'm going to increase my score.

---

### Decision · Program_Chairs · 2021-09-27

**Decision:**

Accept (Poster)

**Comment:**


The reviewers thought this was an interesting paper considering how to combine meta-gradient methods with HRL, and the empirical experiments were convincing. In the internal discussions two reviewers suggested the paper would be further strengthened by a more detailed discussion on the theoretical side. The authors are encouraged to consider the reviewers’ feedback in their revision for the camera ready.